# Constrained Optimization From a Control Perspective via Feedback Linearization

**Runyu Zhang**
Massachusetts Institute of Technology
runyuzha@mit.edu

**Arvind Raghunathan**
Mitsubishi Electric Research Laboratories
raghunathan@merl.com

**Jeff Shamma**
University of Illinois Urbana-Champaign
jshamma@illinois.edu

**Na Li**
Harvard University
nali@seas.harvard.edu

## Abstract

Tools from control and dynamical systems have proven valuable for analyzing and developing optimization methods. In this paper, we establish rigorous theoretical foundations for using feedback linearization (FL)—a well-established nonlinear control technique—to solve constrained optimization problems. For equality-constrained optimization, we establish global convergence rates to first-order Karush-Kuhn-Tucker (KKT) points and uncover the close connection between the FL method and the Sequential Quadratic Programming (SQP) algorithm. Building on this relationship, we extend the FL approach to handle inequality-constrained problems. Furthermore, we introduce a momentum-accelerated feedback linearization algorithm and provide a rigorous convergence guarantee.

## 1 Introduction

Constrained optimization, also known as nonlinear programming, has found vast applications in several domains including robotics [2], supply chains [30], and safe operations of power systems [23]. First-order iterative algorithms are widely used to solve such problems, particularly in optimization and machine learning settings with large-scale datasets. These algorithms can be interpreted as discrete-time dynamical systems, while their continuous-time counterparts, derived by considering infinitesimal step sizes, take the form of differential equations. Analyzing these continuous-time systems can provide valuable theoretical insights, such as stability properties and convergence rates. This perspective is well-developed for unconstrained optimization, exemplified by the gradient flow $\dot{x} = -\nabla f(x)$ [26, 6, 70, 31, 4], the continuous-time counterpart of gradient descent, as well as its accelerated variants [75, 79, 51]. However, for constrained optimization, this approach remains less thoroughly explored.

Recent studies (c.f. [15, 35, 1, 53, 17, 16, 29]) have explored the dynamical properties of continuous time constrained optimization algorithms. These works leverage a feedback control perspective to design and analyze the performance of optimization methods. Specifically, they propose frameworks that model constrained optimization problems as control problems, where the iterations of the optimization algorithm are represented by a dynamical system, and the Lagrange multipliers act as control inputs. The objective in this framework is to drive the system to a feasible steady state that satisfies the constraints. Within this framework, various control strategies can be employed to design the update of Lagrange multipliers, resulting in different control-based first-order methods. For example, it can be shown that Proportional-Integral (PI) control leads to Primal-Dual Gradient Dynamics (PDGD), whose properties are well-studied [46, 66, 22]. However, most of the works focuses on convergence for *convex* constrained problems.

39th Conference on Neural Information Processing Systems (NeurIPS 2025).

In this work, we adopt the same control perspective as above, and specifically focus on using another approach, namely Feedback Linearization (FL) a standard approach in nonlinear control (cf. [45, 41]), to design the Lagrange multiplier. One key advantage of this method is its natural suitability for handling *nonconvex* constrained optimization problems. Although this approach [15], along with similar dynamical system perspectives [72, 1, 52, 54], has been explored in the literature, its theoretical properties are not yet fully understood. Several important questions remain open.

The first question concerns global convergence and convergence rates. While existing works established local stability [15], global convergence and convergence rate have not been established. The second question concerns the relationship between the feedback linearization approach and existing optimization algorithms, specifically whether the discretization of the optimization dynamics derived from feedback linearization aligns with any known optimization method. Additionally, since most existing studies [15, 72] focus exclusively on equality constraints, this raises the third question: how can the feedback linearization approach be extended effectively for inequality constraints? Lastly, it remains unclear whether ideas from acceleration in optimization–such as momentum-based techniques–can be incorporated to speed up feedback lienarization methods for constrained optimization.

**Our contributions.**   Motivated by the open questions discussed above, we aim to deepen the theoretical understanding of the feedback linearization (FL) approach for constrained optimization by addressing these questions. Specifically, our contributions are as follows:

1. We establish a global convergence rate to a first-order Karush-Kuhn-Tucker (KKT) point for the FL method for equality-constrained optimization (Section 3.1).
2. We demonstrate that the FL-based optimization algorithm is closely related to the Sequential Quadratic Programming (SQP) algorithm, providing a new perspective on its connection to established optimization techniques (Section 3.2).
3. Building on this insight, we extend the method to handle inequality constraints, broadening its applicability (Section 4).
4. Finally, we propose a momentum-accelerated FL algorithm for constrained optimization, which empirically exhibits accelerated convergence in both equality constrained and inequality constrained settings. Furthermore, we establish $O(\frac{1}{\sqrt{T}})$ convergence in the nonconvex equality constrained setting (Section 5).

Due to space limits, a comprehensive review of related literature is deferred to Appendix A.

**Notations:**   We use the notation $[n], n \in \mathbb{N}$ to denote the set $\{1, 2, 3, \ldots, n\}$. We use $\nabla f(x)$ to denote the gradient of a scalar function $f : \mathbb{R}^n \to \mathbb{R}$ evaluated at the point $x \in \mathbb{R}^n$ and use $\nabla^2 f(x)$ to denote its corresponding Hessian matrix. We use $J_h(x)$ to denote the Jacobian matrix of a function $h : \mathbb{R}^n \to \mathbb{R}^m$ evaluated at $x \in \mathbb{R}^n$, i.e. $[J_h(x)]_{i,j} = \frac{\partial h_i(x)}{\partial x_j}$, $i \in [m], j \in [n]$. Unless specified otherwise, we use $\| \cdot \|$ to denote the $L_2$ norm of matrices and vectors and use $\| \cdot \|_\infty$ to denote the $L_\infty$ norm. For a positive definite matrix $A$, we use $\|X\|_A := \|A^{-\frac{1}{2}} X\|$ to denote the $A$-norm of $X$. For a set $\mathcal{A}$, we use $\mathcal{A}^c$ to denote its complement. When no ambiguity arises, we denote $\frac{dx}{dt}$ by $\dot{x}$, and abbreviate time-dependent variables such as $x(t)$, $\lambda(t)$, and $y(t)$ as $x$, $\lambda$, and $y$.

## 2   Feedback Linearization (FL) for solving equality constrained optimization

In this section, we briefly review related works that adopt a control perspective, particularly focusing on the use of feedback linearization (FL) to address equality-constrained optimization problems.

**Control perspective on equality-constrained optimization [15]**   Consider the constrained optimization problem with equality constraints

$$\min_x f(x) \qquad s.t. \ h(x) = 0, \tag{1}$$

where $x \in \mathbb{R}^n$, $f : \mathbb{R}^n \to \mathbb{R}, h : \mathbb{R}^n \to \mathbb{R}^m$.   Here we assume that $f, h$ are differentiable, and additional assumptions will be introduced where needed to support the analysis. The first-order KKT conditions are given by

$$-\nabla f(x) - J_h(x)^\top \lambda = 0, \quad h(x) = 0 \tag{2}$$

The key idea is to view finding the KKT point as a control problem (Figure 1) with the system dynamics given by,

$$\frac{dx}{dt} = -T(x(t))\left(\nabla f(x(t)) + J_h(x(t))^\top \lambda(t)\right),$$
$$y(t) = h(x(t)),$$

(3)

where $x$ represents the system state, $y = h(x)$ is system constraint variable and $\lambda$ is the control input. $T(x)$ here is a positive definite matrix and throughout the paper we assume that there exists $\lambda_{\min}, \lambda_{\max}$ such that for all $x$,

$$\lambda_{\min} I \preceq T(x) \preceq \lambda_{\max} I$$

Note that at an equilibrium point $x^\star$ of the system in Fig. 1 must satisfy: $\dot{x} = 0 \implies \nabla f(x^\star) + J_h^\top(x^\star)\lambda = 0$.

Further, if $x^\star$ is feasible, i.e. $h(x^\star) = 0$, then we get that $x^\star$ satisfies the first order KKT conditions (2). Thus, the key idea is to manipulate the evolution of $x$ so that we stabilize the system to equilibrium and feasibility.

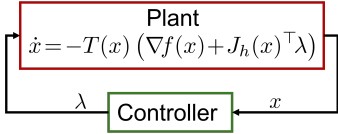

Figure 1: Control Perspective for Constrained Optimization

To measure convergence, We define the KKT-gap of $(x, \lambda)$ as follows, such that converging to a KKT point is equivalent to KKT-gap is zero [1]:

$$\texttt{KKT-gap}(x, \lambda) := \max\{\|\nabla f(x) + J_h(x)^\top \lambda\|, \|h(x)\|_\infty\}.$$

(4)

To design the controller $\lambda(t)$ to reach a feasible equilibrium, we next introduce the feedback linearization (FL) approach, which is the main focus of this paper.

**Feedback linearization (FL) for equality-constrained optimization [15]**   Feedback linearization (FL) [45, 41] is a classical control method for controlling nonlinear dynamics which generally takes the following form:

$$\dot{x} = F(x) + G(x)\lambda$$

(5)

Directly designing a stabilizing controller for the nonlinear system is a challenging task. The FL approach circumvents the difficulty by transforming the nonlinear control problem into an equivalent linear control problem, which is much easier to analyze, through a change of variables and a suitable control input. In particular, if $G(x)$ is invertible, then one can let $\lambda := G(x)^{-1}(u - F(x))$ where $u(t)$ is the new control input to be designed. Substituting $\lambda = G(x)^{-1}(u - F(x))$ into the dynamics (5), the dynamics becomes $\dot{x} = u$, which is now a linear system.

Similarly, in the equality constrained optimization problem, we write out the dynamics for $y$:

$$\dot{y} = J_h(x)\dot{x} = \underbrace{-J_h(x)T(x)}_{F(x)} \nabla f(x) \underbrace{-J_h(x)T(x)J_h(x)^\top}_{G(x)} \lambda$$

Thus, by setting

$$\lambda = -\left(J_h(x)T(x)J_h(x)^\top\right)^{-1}(u + J_h(x)T(x)\nabla f(x))$$

we have that $\dot{y} = u$, then we can simply set $u = -Ky$ where $K$ is a Hurwitz matrix to guarantee that $y$ asymptotically converge to zero. Thus the feedback linearization (FL) dynamics is given as:

---

**FL for Equality-Constrained Optimization [15]**

$$\dot{x} = -T(x)\left(\nabla f(x) + J_h(x)^\top \lambda\right)$$
$$\lambda = -\left(J_h(x)T(x)J_h(x)^\top\right)^{-1}\left(J_h(x)T(x)\nabla f(x) - Kh(x)\right)$$

(6)

---

The FL approach is particularly effective for handling nonlinear dynamics, making it well-suited for nonconvex constrained optimization. Numerical results in [15, 72] highlight its strong performance

---

[1]In this paper, we use the term 'convergence' to refer to the decay of the KKT gap to zero. While this is weaker than convergence to a local or global optimum, it remains a meaningful guarantee in nonconvex constrained optimization. Extending the framework to incorporate saddle-point escape techniques and ensure convergence to local optima is an important direction for future work.

in such settings. However, its theoretical properties remain less well understood. Existing analyses primarily focus on local stability [15], while global convergence and convergence rates are largely unexplored. Additionally, the connection between the FL algorithm and existing optimization methods is not well established. It also remains unclear how to leverage the FL approach to develop novel techniques for inequality-constrained optimization and faster constrained optimization. In the following sections, we will systematically address these open problems.

**Remark 1** (Scalability and Computational Complexity). *Note that although* (6) *requires calculating the matrix inversion of* $J_h(x)T(x)J_h(x)^\top \in \mathbb{R}^{m \times m}$*, the dimension scales with the number of constraints* $m$ *instead of the dimension of the optimization variable* $x \in \mathbb{R}^n$*. In many practical settings, e.g. safe RL, the number of constraints is significantly smaller than the dimension of* $x$*. In such cases, the inversion is computationally inexpensive and can be performed efficiently. Moreover, even in cases where the number of constraints is large and the matrix inversion becomes computationally burdensome, we show that it is possible to approximate the inverse efficiently while still maintaining convergence guarantees. Specifically, in Appendix B.2, we present a modified FL algorithm that incorporates a Proportional-Integral (PI) controller to tolerate approximation error and still ensure convergence to an optimal solution.*

**Remark 2** (Extensions of the FL approach). *Although similar dynamics to* (6) *has also been proposed from other perspectives such as control barrier functions [1], nonsmooth dynamics design [53, 55], we believe that the feedback linearization perspective provides a more general framework. In principle, beyond specifying a linear target of the form* $\dot{y} = -Ky$*, this approach allows us to leverage arbitrary stable controllers, such as PI controllers, e.g.*

$$\dot{y} = -K_p y - K_i \int_0^t y(s)ds \tag{7}$$

*for constraint enforcement, potentially offering new algorithmic behaviors or robustness benefits. In Appendix B.2, we present an example where the algorithm in* (7) *succeeds in converging to the optimal solution, even when only an inaccurate approximation of* $\left(J_h(x)T(x)J_h(x)^\top\right)^{-1}$ *is available—while* (6) *fails to converge to an optimal solution. This highlights the robustness and broader applicability of the feedback linearization framework.*

## 3 FL control method: Convergence and Relationship to SQP

### 3.1 Convergence Results

Section 2 introduces the FL method for equality-constrained optimization. The analyses in existing works mainly focus on the local stability, and little is known about the global convergence property. In this section, we establish a global convergence rate to a first order KKT point (Contribution 1).

The result relies on the following assumptions:

**Assumption 1.** *There exists a constant $M$ such that $\|\nabla f(x)\| < M$, $\|J_h(x)\| < M$ for all $x$;*

**Assumption 2.** *The function $f(x)$ is lower-bounded, i.e. $f(x) \geq f_{\min}$ for all $x$.*

**Assumption 3.** *There exists a constant $D$ such that $(J_h(x)J_h(x)^\top)^{-1} \prec D^2 I$ for all $x$.*

Note that Assumption 3 is similar to the assumption made in [15] that assumes that $\text{rank}(J_h(x)) = m$ for all $x$, which is equivalent to $J_h(x)J_h(x)^\top$ being invertible, thereby ensuring the regularity of the transformation in (6) and the existence of a well-defined feedback linearization. This assumption is also known as the linear independence constraint qualification (LICQ, cf. [61, 57], see more discussion in Appendix A) in optimization literature. Assumption 1 implies that the functions $f$ and $g$ are Lipschitz. We would like to acknowledge that this assumption is relatively restrictive and is solely for analysis purpose [2]. In our numerical simulations we found that the algorithm is suitable for non-uniformly-Lipschitz functions. We now state our result in terms of the convergence rate:

**Theorem 1.** *Let Assumption 1, 2 and 3 hold and let the control gain $K$ be a diagonal positive definite matrix, i.e., $K = \text{diag}\{k_i\}_{i=1}^m$, where $k_i > 0$. Then we have that the dynamic of the feedback linearization method* (6) *satisfies:*

---

[2]We note that if $x^\star \in D$ is known a priori for a compact domain $D$, a potential approach for handling non-uniformly Lipschitz functions $f, g$ is to construct Lipschitz extensions $f', g'$ such that their gradients and Jacobians match those of $f, g$ within $D$ while remaining uniformly Lipschitz outside $D$ (cf. [74]).

1. *For the set $\mathcal{E}_i := \{x : h_i(x) \geq 0\}$, if $x(0) \in \mathcal{E}_i$, then $x(t) \in \mathcal{E}_i$ for all $t \geq 0$. Similarly, if $x(0) \in \mathcal{E}_i^c$, then $x(t) \in \mathcal{E}_i^c$ for all $t \geq 0$, further $h_i(x(t)) = e^{-k_i t} h_i(x(0))$, i.e., $h(x(t)) \to 0$ with an exponential rate as $t \to +\infty$.*

2. *Define $\ell(x) := f(x) + \frac{\lambda_{\max}}{\lambda_{\min}}(MD)^2 \sum_{i=1}^m |h_i(x)|$, then $\ell(x(t))$ is non-increasing w.r.t. $t$.*

3. *Let $\overline{\lambda}(t) := -\left(J_h(x)T(x)J_h(x)^\top\right)^{-1} J_h(x)T(x)\nabla f(x(t))$, then we have that*

$$\int_{t=0}^T \|\nabla f(x(t)) + J_h(x(t))^\top \overline{\lambda}(t)\|^2 dt \leq \frac{1}{\lambda_{\min}}\left(\ell(x(0)) - \ell(x(T))\right),$$

   *and that $\lim_{t\to+\infty}\left(\lambda(t) - \overline{\lambda}(t)\right) = 0$.*

4. *(Asymptotic convergence and convergence rate) The above statements imply that,*

$$\inf_{0\leq t\leq T} \texttt{KKT-gap}(x(t), \overline{\lambda}(t)) \leq \max\left\{ \sqrt{\frac{2}{T}\left(\frac{f(x(0))-f_{\min}}{\lambda_{\min}} + \frac{\lambda_{\max} M^2 D^2}{\lambda_{\min}^2}\sum_i |h_i(x(0))|\right)},\right.$$
$$\left. \max_{1\leq i\leq m}\left\{h_i(x(0))e^{-\frac{k_i T}{2}}\right\}\right\} \sim O\left(\frac{1}{\sqrt{T}}\right)$$

   *further, we have that $\lim_{t\to+\infty}\texttt{KKT-gap}(x(t), \overline{\lambda}(t)) = 0$, $\lim_{t\to+\infty}\texttt{KKT-gap}(x(t), \lambda(t)) = 0$.*

Statement 4 in Theorem 1 implies that the algorithm can find an $\epsilon$-first-order-KKT-point within time $\frac{1}{\epsilon^2}$. We note that ensuring last-iterate convergence in nonconvex optimization is generally challenging. Hence, our analysis focuses on the best iterate, a widely adopted criterion in nonconvex optimization. However, in the setting where the optimization problem (1) is strongly convex, we are able to strengthen our convergence result to the last iterate convergence to the global optimal solution. Due to space limitations, we defer the detailed proof of Theorem 1 as well as the result for the strongly convex setting to Appendix C. The key step of the proof involves constructing the merit function $\ell(x)$ in Statement 2. We also note that $\ell(x)$ also serves as the exact penalty function in constrained optimization literature (cf. [27, 84]).

## 3.2 Relationship with SQP

The FL dynamics (6) provides a concise and elegant formulation, prompting the question of whether certain optimization algorithms can be derived through its discretization. In this section, we establish a fundamental connection between the continuous-time FL dynamics and the Sequential Quadratic Programming (SQP) algorithm (Contribution 2). Specifically, we demonstrate that the forward-Euler discretization (cf. [8, 7]) of (6) is equivalent to the SQP algorithm.

The state space continuous time dynamic for (6) is

$$\dot{x} = -T(x)\left(\nabla f(x) - J_h(x)^\top \left(J_h(x)T(x)J_h(x)^\top\right)^{-1}\left(J_h(x)T(x)\nabla f(x) - Kh(x)\right)\right).$$

Its forward-Euler discretization scheme is

$$x_{t+1} = x_t - \eta T(x_t)\left(\nabla f(x_t) - J_h(x_t)^\top\left(J_h(x_t)T(x_t)J_h(x_t)^\top\right)^{-1}\left(J_h(x_t)T(x_t)\nabla f(x_t) - Kh(x_t)\right)\right) \tag{8}$$

We now consider the following SQP method, which is widely discussed in literature (cf. [57, 12, 58]):

$$x_{t+1} = \arg\min_x \nabla f(x_t)^\top (x - x_t) + \frac{1}{2\eta}(x - x_t)^\top T(x_t)^{-1}(x - x_t)$$
$$s.t. \quad h(x_t) + J_h(x_t)(x - x_t) = 0 \tag{9}$$

We are now ready to state the main result of this section, which demonstrates the equivalence of (8) and (9)

**Theorem 2.** *Under Assumption 3, when $K = \frac{1}{\eta}I$, the discretization of FL (8) is equivalent to the SQP algorithm (9).*

The proof of Theorem 2 leverages the fact that (9) satisfies the relaxed Slater condition. The detailed proof is deferred to Appendix D.

**Remark 3** (Choice of $T(x)$). *Theorem 2 provides insights into the selection of $T(x)$ for the FL approach. Different choices of $T(x)$ lead to different types of SQP algorithms. Here we discuss two specific types of $T(x)$. First, $T(x)$ is set as the inverse of the Hessian matrix, i.e., $T(x) = \left(\nabla^2 f(x)\right)^{-1}$, then (9) corresponds to the Newton-type algorithm where the quadratic term in the objective function is given by $(x - x_t)^\top \nabla^2 f(x)(x - x_t)$, which is widely considered in literature (cf. [57, 12]). For this specific type of $T(x)$, we name its corresponding FL dynamics (6) as the* **FL-Newton** *method. However, in the setting where the Hessian information is not available, another choice of $T(x)$ is simply setting it as the identity matrix $T(x) = I$, which is considered in recent works such as [58]. In this case, the objective function resembles a proximal operator (cf. [13, 60]), hence we name this as* **FL-proximal** *method. Due to space limit, we defer a more comprehensive overview of SQP to Appendix A. We would also like to emphasize that FL-proximal belongs to the class of first-order methods as its update only requires the first-order information $\nabla f(x), J_h(x)$.*

**Remark 4** (Comparison with other first-order methods). *To this end, we briefly compare FL-proximal methods with other first-order approaches, including Primal-Dual Gradient Descent (PDGD), Projected Gradient Descent (PGD), and the Augmented Lagrangian Method (ALM), with further details in Appendix A. PDGD has been well studied [46, 77, 66], but is largely limited to convex settings and may fail in nonconvex problems [87, 5]. PGD also struggles with nonconvexity, as projections onto nonconvex sets are often intractable. ALM can handle nonconvex constraints, but each iteration requires solving a potentially expensive nonconvex subproblem. In contrast, when the constraint dimension is small relative to that of $x$, SQP-based methods like ours often perform better in practice [34, 49].*

## 4   Extension to inequality constraints

The above sections primarily focus on the constrained optimization setting with equality constraints (1). This section aims to address the question of whether we can extend to setting with inequality constraints (Contribution 3), i.e.,

$$\min_x f(x) \qquad s.t. \ h(x) \leq 0, \tag{10}$$

The KKT conditions for the above problem are given by

$$-\nabla f(x) - J_h(x)^\top \lambda = 0, \quad h(x) \leq 0, \quad \lambda \geq 0, \quad \lambda^\top h(x) = 0 \tag{11}$$

To measure convergence, we define the KKT-gap of the state variable $x$ and nonnegative control variable $\lambda \geq 0$ as follows:

$$\texttt{KKT-gap}(x, \lambda) := \max \left\{ \|\nabla f(x) + J_h(x)^\top \lambda\|, \left|\lambda^\top h(x)\right|, \max_i [h_i(x)]_+ \right\},$$

where $[h_i(x)]_+ = \max\{h_i(x), 0\}$.

We can still view the problem as a control problem whose corresponding dynamics can be written as

$$\dot{x} = -T(x)\left(\nabla f(x) + J_h(x)^\top \lambda\right), \quad y = h(x), \quad \lambda \geq 0. \tag{12}$$

However, the problem becomes more complicated because we require the non-negativity constraints $\lambda \geq 0$ and complementary slackness $\lambda^\top h(x) = 0$. At first glance, it is unclear how to guarantee these conditions through the control process. However, inspired by the relationship with SQP algorithms, we carefully design a more intricate FL controller as follows:

---
**FL for Inequality-Constrained Optimization**

$$\dot{x} = -T(x)\left(\nabla f(x) + J_h(x)^\top \lambda\right) \tag{13.1}$$

$$\lambda = \arg\min_{\lambda \geq 0} \left(\tfrac{1}{2}\lambda^\top J_h(x)T(x)J_h(x)^\top \lambda + \lambda^\top \left(J_h(x)T(x)\nabla f(x) - Kh(x)\right)\right) \tag{13.2}$$
---

Here we assume that the optimization problem in equation (13.2) admits a finite solution. We would also like to point out that $\lambda$ in (13.2) takes the form of the solution of an optimization problem, resulting in a non-smooth trajectory. A similar formulation of non-smooth ordinary differential equations (ODEs) has been explored in the context of differential variational inequalities (cf. [24, 59, 14]).

At first glance, it may not be immediately clear why the algorithm is structured as in (13). The derivation of (13) was inspired by the connection between the FL method and SQP in the equality-constrained setting. Hence, for the inequality-constrained case, we first analyzed SQP and then reverse-engineered its principles to derive its continuous-time counterpart, leading to the formulation of the FL method in (13). To ensure a coherent and intuitive presentation, we begin by establishing its relationship with the SQP algorithm.

**Relationship with the SQP algorithm**   The corresponding forward Euler discretization of (13) is given by

$$
\begin{aligned}
x_{t+1} &= x_t - \eta T(x)\left(\nabla f(x_t) + J_h(x_t)^\top \lambda_t\right) \\
\lambda_t &= \arg\min_{\lambda \geq 0}\left(\tfrac{1}{2}\lambda^\top J_h(x_t)T(x_t)J_h(x_t)^\top \lambda + \lambda^\top\left(J_h(x_t)T(x_t)\nabla f(x_t) - Kh(x_t)\right)\right)
\end{aligned}
\tag{14}
$$

We now consider the following SQP type of optimization method

$$
\begin{aligned}
x_{t+1} = \arg\min_x \nabla f(x_t)^\top(x - x_t) + \tfrac{1}{2\eta}(x - x_t)^\top T(x_t)^{-1}(x - x_t) \\
s.t. \quad h(x_t) + J_h(x_t)(x - x_t) \leq 0
\end{aligned}
\tag{15}
$$

The following theorem states the equivalence between (14) and (15).

**Theorem 3.** *When $K = \frac{1}{\eta}I$, if (15) is feasible, then the discretization of feedback linearization (14) is equivalent to the SQP algorithm (15).*

Similar to the proof of Theorem 2, the proof of Theorem 3 also leverages strong duality and KKT conditions. The detailed proof is deferred to Appendix D.

**Convergence Result**   Theorem 3 demonstrates the relationship between the FL algorithm (13) and (15). Since SQP algorithms are known to be capable of converging to a KKT point [57], intuitively similar convergence can be established for our FL algorithm (13), which is the main focus of the following part.

We define the index set $\mathcal{I}(x) := \{i : h_i(x) > 0\}$. Our results rely on the following assumptions:

**Assumption 4.** *Given the initial state $x(0)$ at $t = 0$, the optimization problem in (13.2) admits a bounded solution $\|\lambda\|_\infty \leq L$ for all $x \in \mathcal{E}$, where $\mathcal{E}$ is defined by $\mathcal{E} := \{x | 0 < h_i(x) \leq h_i(x(0)), \forall i \in \mathcal{I}(x(0))\}$.*

Although Assumption 4 is quite complicated, there are some simplified versions that serve as a sufficient condition of Assumption 4. For example, if we start with a feasible $x(0)$, then $\mathcal{E} = \emptyset$ and hence Assumption 4 is automatically satisfied. Additionally, note that Assumption 3 is another sufficient condition of Assumption 4 (see Lemma 3 in Appendix G). Notably Assumption 4 is similar to the Mangasarian-Fromovitz constraint qualification (MFCQ) considered in literature [53, 1]

**Theorem 4.** *Let Assumption 1, 2 and 4 hold and let the control gain matrix $K$ be a diagonal matrix, i.e., $K = \mathrm{diag}\{k_i\}_{i=1}$, where $k_i > 0$. Then the learning dynamics (13) satisfies the following properties*

1. *$\frac{dh_i(x(t))}{dt} \leq -k_i h_i(x(t))$, and hence the dynamic will converge to the feasible set.*

2. *Define $\ell(x) := f(x(t)) + L\sum_i [h_i(x)]_+$, then $\ell(x(t))$ is non-increasing w.r.t $t$. Here $[h_i(x)]_+ = \max\{h_i(x), 0\}$.*

3. *The following inequality holds*

$$
\int_{t=0}^{T}\left(\|\nabla f(x(t)) + J_h(x(t))\lambda(t)\|_{T(x(t))}^2 - \sum_{i \in \mathcal{I}(x)^c} k_i \lambda_i(t) h_i(x(t))\right) dt \leq \ell(x(0)) - \ell(x(T))
$$

4. *(Asymptotic convergence and convergence rate) The above statements imply that*

$$
\inf_{0 \leq t \leq T} \texttt{KKT-gap}(x(t), \lambda(t)) \leq \max\left\{\sqrt{\tfrac{2}{\lambda_{\min}T}\left(f(x(0)) - f_{\min} + L\sum_{i \in \mathcal{I}(x(0))} h_i(x(0))\right)},\right.
$$

$$
\left.\tfrac{1}{\min_i k_i}\tfrac{2}{T}\left(f(x(0)) - f_{\min} + (L+1)\sum_{i \in \mathcal{I}(x(0))} h_i(x(0))\right)\right\} \sim O\left(\tfrac{1}{\sqrt{T}}\right)
$$

   *Further    we    have    that    `KKT-gap`    asymptotically    converges    to    zero,    i.e. $\lim_{t \to +\infty} \texttt{KKT-gap}(x(t), \lambda(t)) = 0$*

Statement 4 in Theorem 4 implies that the algorithm can find an $\epsilon$-first-order KKT-point within time $\frac{1}{\epsilon^2}$. Similar to Theorem 1, the key step of the proof is to construct the merit function in Statement 2 (detailed proof deferred to Appendix E).

## 5  Momentum Acceleration for Constrained Optimization

In Remark 3, we introduced the FL-proximal and FL-Newton algorithms. Generally, FL-Newton achieves faster convergence than FL-proximal due to its use of second-order information. However, in scenarios where Hessian information is unavailable, FL-proximal must be used instead, raising the question of whether its convergence can be accelerated. Given that momentum acceleration has been shown to improve convergence rates in unconstrained optimization, a natural question arises: can a momentum-accelerated version of the FL-proximal algorithm, along with its corresponding discrete-time SQP formulation, achieve faster convergence? This section aims to address this question as part of Contribution 4.

Momentum acceleration is a technique commonly used in optimization to enhance convergence rates (cf. [63, 56, 20], see Appendix A for more detailed introduction about momentum acceleration). For unconstrained optimization, the discrete-time momentum acceleration for gradient descent generally takes the form of

$$w_t = x_t + \beta(x_t - x_{t-1}), \quad x_{t+1} = w_t - \eta\nabla f(w_t) \tag{16}$$

Its corresponding continuous-time analogue can be written as a second-order ODE [63, 75]

$$\dot{x} = z, \quad \dot{z} = -\alpha z - \nabla f(x) \tag{17}$$

Inspired by the form of (16) and (17), for equality constrained optimization, we propose the following heuristic momentum-accelerated discrete time SQP scheme

$$w_t = x_t + \beta(x_t - x_{t-1}), \quad x_{t+1} = w_t + \eta\nabla f(w_t) + J_h(w_t)^\top \lambda_t$$
$$\lambda_t = -\left(J_h(w_t)J_h(w_t)^\top\right)^{-1}\left(J_h(w_t)\nabla f(w_t) - \frac{1}{\eta}h(w_t)\right) \tag{18}$$

and continuous time FL scheme, which we name as **FL-momentum**:

---
**FL-momentum for Equality-Constrained Optimization**

$$\dot{x} = z \qquad \dot{z} = -\alpha z - \left(\nabla f(x) + J_h(x)^\top \lambda\right)$$
$$\lambda = -(J_h(x)J_h(x)^\top)^{-1}(J_h(x)\nabla f(x) - Kh(x)) \tag{19}$$

---

Note that compared with FL-proximal (8), the difference in (18) is the addition of a momentum step $w_t = x_t + \beta(x_t - x_{t-1})$. Similarly we can propose the FL-momentum scheme for inequality constraint case as follows:

---
**FL-momentum for Inquality-Constrained Optimization**

$$\dot{x} = z, \qquad \dot{z} = -\alpha z - \left(\nabla f(x) + J_h(x)^\top \lambda\right) \tag{20}$$
$$\lambda = \arg\min_{\lambda \geq 0} \frac{1}{2}\lambda^\top J_h(x)J_h(x)^\top \lambda + \lambda^\top \left(J_h(x)\nabla f(x) - Kh(x)\right)$$

---

The numerical simulation in Section 6 (Figure 2) suggests that momentum methods indeed accelerate the convergence rate. We would also like to note that as far as we know, the acceleration of SQP methods are generally achieved via Newton or quasi-Newton methods, there's little work on exploring acceleration via momentum approaches, which makes our proposed momentum algorithm a novel contribution.

### 5.1  Convergence Analysis for FL-momentum: Nonconvex Case

In this section, we provide some convergence guarantees for the proposed algorithm. In particular, we primarily focus on the convergence analysis for the continuous-time algorithm for equality constrained optimization (19). It remains future work to establish the convergence for the discrete-time algorithm (18) or the inequality-constrained algorithm (20).

We first define the following notation

$$\overline{\lambda}(x) := -(J_h(x)J_h(x)^\top)^{-1}(J_h(x)\nabla f(x)) \tag{21}$$

Apart from Assumption 1 and 2, we also make the following assumptions on $f$ and $h$.

**Assumption 5.** *Both $f(x)$, $h(x)$ are three-times differentiable and the derivatives are bounded, thus, we know that there exist some constants $L_f, L_1, L_2$ such that*

$$\|\nabla^2 f(x)\| \le L_f, \quad \left\|\frac{\partial \bar{\lambda}(x)}{\partial x}\right\| \le L_1, \quad \left\|\frac{\partial\left(J_h(x)^\top \lambda(x) + \left(\frac{\partial \bar{\lambda}(x)}{\partial x}^\top h(x)\right)\right)}{\partial x}\right\| \le L_2$$

**Assumption 6.** *We also assume that that there exists a constant $\bar{H}$ such that $\|\bar{H}(x)\| \le \bar{H}, \ \forall x$, where $\bar{H}(x) := [h(x)^\top \nabla^2 h_i(x)]_{i=1}^n$.*

We are now ready to state our main result

**Theorem 5.** *Assume that Assumption 1, 2, 5 and 6 hold. Let two positive constants $a_1, a_2$ be such that $a_2 \ge \left(4\frac{\lambda_{\max}(K)}{\lambda_{\min}(K)}L_2 D + \frac{L_1^2}{\lambda_{\min}(K)}\right) \times a_1 \ge 0$. We define the following merit function:*

$$\ell(x, z) = a_1 \alpha f(x) + \frac{a_2 \alpha}{2}\|h(x)\|^2 + a_1 \alpha \bar{\lambda}(x)^\top h(x) + \|z\|^2$$

$$+ \left(a_1 \nabla f(x) + a_2 J_h(x)^\top h(x) + a_1 J_h(x)^\top \bar{\lambda}(x) + a_1 \frac{\partial \bar{\lambda}(x)}{\partial x}^\top h(x)\right)^\top z$$

*then for $\alpha \ge \left(a_1(L_f + L_2) + a_2(M^2 + \bar{H}) + \frac{1}{a_1} + \frac{2(\lambda_{\max}(K)D^2)}{a_2}\right) + 1$, we have that*

1. *$\ell(x(t), z(t))$ is non-increasing with respect to $t$.*

2. *the following inequality holds*

$$\int_{t=0}^{T} \frac{a_2 \lambda_{\min}(K)}{8}\|h(x(t))\|^2 + \frac{a_1}{4}\|\nabla f(x(t)) + J_h(x(t))^\top \bar{\lambda}(x(t))\|^2 dt \le \ell(x(0), z(0)) - \min_{x,z}\ell(x, z)$$

3. *We can bound the KKT-gap by*

$$\inf_{0 \le t \le T} \texttt{KKT-gap}(x(t), \bar{\lambda}(x(t))) \le \sqrt{\frac{\ell(x(0),z(0)) - \ell_{\min}}{\min\left\{\frac{a_2\lambda_{\min}(K)}{8}, \frac{a_1}{4}\right\}T}} \sim O(\tfrac{1}{\sqrt{T}})$$

*and $\lim_{t\to+\infty} \texttt{KKT-gap}(x(t), \bar{\lambda}(x(t))) = 0, \quad \lim_{t\to+\infty} \texttt{KKT-gap}(x(t), \lambda(x(t))) = 0$.*

The detailed proof is provided in Appendix F.

**Remark 5** (Limitation of the result). *One limitation of Theorem 5 is that it establishes convergence but not acceleration over FL-proximal. However, when the constraint function $h(x)$ is affine, the algorithm is equivalent to the momentum-accelerated projected gradient method (see Appendix F.1), offering insight into its potential for accelerating optimization.*

## 6 Numerical Verifications

For numerical validation, we consider a logistic regression problem involving heterogeneous clients [73, 42]. Many scenarios, such as federated learning and fair machine learning, require training a common model in a distributed manner by utilizing data samples from diverse clients or distributions. In practice, heterogeneity in local data distributions often results in uneven model performance across clients [48, 76]. Since this outcome may be undesirable, a reasonable objective in such settings is to add constraints to ensure that the model's loss is comparable across all clients.

We formulate the above problem as a constrained optimization problem as follows: consider solving the logistic regression for $C$ clients. For each client $c \in \{1, 2, \ldots, C\}$, it is associated with its own dataset $D_c = \{(x_i, y_i)\}_{i=1}^{|D_c|}$, where the label is $y_i \in \{-1, 1\}$ and data feature is $x_i \in R^d$. For each client $c$, its own logistic regression loss $R_c(\theta)$ is defined as

$$R_c(\theta) := \frac{1}{|D_c|}\sum_{i \in D_c}\log(1 + \exp(-y_i \cdot \theta^\top x_i)),$$

where $\theta$ is the parameter of the regression model. We further define the averaged regression loss $\bar{R}(\theta)$ as $\bar{R}(\theta) := \frac{1}{C}\sum_{c=1}^{C}f_c(\theta)$.

As suggested in [73, 42], heterogeneity challenges can be addressed by introducing a proximity constraint that links the performance of each individual client, $R_c$, to the average loss across all clients, $\bar{R}$. This approach naturally formulates a constrained learning problem [3]

$$\min_\theta \bar{R}(\theta), \quad s.t.\ R_c(\theta) - \bar{R}(\theta) - \epsilon \leq 0,\ \forall c \in \{1, 2, \ldots, C\} \tag{22}$$

where $\epsilon > 0$ is a small, fixed positive scalar.

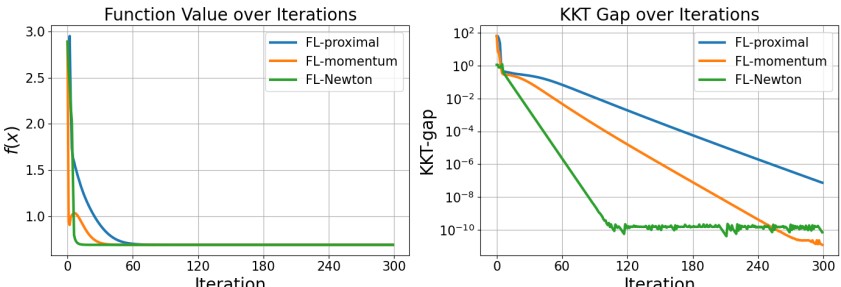

Figure 2: Result for Heterogeneous Logistic Regression

We solve the constrained optimization problem (22) by running the FL-proximal, FL-Newton, and FL-momentum algorithm . Here we set the number of clients to $C = 5$ and $|D_c| = 200$, the data $y_i$ is randomly generated from a Bernoulli distribution and $x_i$ is generated from a Gaussian distribution whose mean differs among different agents. The results of the numerical simulation are presented in Figure 2. All algorithms converge to a first-order KKT point. FL-Newton converges fastest owing to its use of second-order (Hessian) information, while among first-order methods, FL-momentum outperforms FL-proximal in convergence speed. More comprehensive numerical results are provided in Appendix B.

# 7 Conclusion

In this paper, we study the theoretical foundations for solving constrained optimization problems from a control perspective via feedback linearization (FL). We established global convergence rates for equality-constrained optimization, highlighted the relationship between FL and Sequential Quadratic Programming (SQP), and extended FL methods to handle inequality constraints. Furthermore, we introduced a momentum-accelerated FL algorithm, which empirically demonstrated faster convergence and provided rigorous convergence guarantees for its continuous-time dynamics. Future directions include exploring the potential extension to zeroth-order optimization settings and relaxing assumptions in the theoretical analysis.

Several limitations of our framework remain. First, our analysis ensures convergence of the best-found iterate to a first-order KKT point, but does not distinguish between local optima, global optima, and saddle points. Extending the framework to guarantee convergence to local optima via saddle-point escape mechanisms is an important direction for future work. Second, the applicability of feedback linearization depends on structural assumptions such as the Linear Independence Constraint Qualification (LICQ). Third, while FL-momentum empirically accelerates convergence, our current analysis yields the same $O(1/\sqrt{T})$ rate as its non-accelerated counterpart. Relaxing these assumptions and strengthening convergence guarantees remain promising directions for future research.

## Acknowledgement

This work was supported in part by NSF ASCENT under Grant 2328241 and in part by the NSF AI Institute under Grant 2112085. Runyu Zhang was supported by the MIT Postdoctoral Fellowship Program for Engineering Excellence.

---

[3]It is not hard to verify that with probability one Equation (22) satisfies Assumption 2 and 4. Also within a compact region, the function satisfies Assumption 1 (Lipschitz property) as well

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

# A Related Literature

**Control and Dynamical System Perspective for Optimization**  As mentioned in the introduction, many papers analyze the performance continuous time optimization algorithms in different settings. For unconstrained cases, various continuous algorithms are studied such as gradient flow [26, 6, 70, 31, 4], stochastic gradient [67, 82], Nesterov acceleration [75, 79, 51], and momentum acceleration [63, 62]. Notably, control-theoretic tools, such as integral quadratic constraints (IQC) [47, 44], Proportional Integral Derivative (PID) control [43], are used to bring interesting insights to the convergence property of optimization algorithms. However, for constrained cases, less work is known. One direction focuses on the dynamics of PDGD algorithm, which primarily focuses on the convex setting and establishing global convergence [46, 77, 22, 85, 15]. However, this approach is primarily limited to convex optimization. For nonconvex optimization, prior research has explored modifications of the gradient flow to accommodate equality constraints (see, e.g., [83, 72, 78]) as well as inequality constraints [1, 52, 54] . The algorithms proposed in these works share similarities with the feedback linearization approach considered in [15] and this paper. Notably, [72] also highlighted connections to sequential quadratic programming (SQP). This paper advances the existing literature in several key ways: i) while prior analyses [83, 72, 89] primarily focus on equality constraints, our results extend the algorithm to handle inequality constraints; ii) although the algorithms in these works resemble ours, we introduce a novel perspective based on feedback linearization; iii) previous convergence results are largely asymptotic, whereas we establish non-asymptotic convergence rates in terms of the KKT gap; and iv) We establish acceleration results through the momentum method, further contributing to the theoretical understanding of the algorithm. Notably, [54] also considers a momentum-based acceleration for constrained optimization, however, both the continuous-time dynamics and the discretization approach are different, and thus resulting in distinct algorithms.

Another related line of research explores real-time optimization from a systems perspective [25, 38, 39, 40, 88]. These works primarily focus on scenarios where the optimizer (controller) interacts with a physical dynamical system (plant) and employ control-theoretic approaches, such as time-scale separation and the small-gain theorem, to design and analyze the performance of real-time interactive optimization schemes. In contrast, our setting differs in that the plant itself corresponds to the gradient dynamics of the Lagrangian, while the controller is represented by the dual variable $\lambda$.

A series of works have also explored the properties of learning and optimization in neural networks using tools from control and dynamical systems, including the proportional-integral-derivative (PID) approach [3], quadratic constraints [28], and Lyapunov stability [68]. However, as these studies focus on different problem settings and techniques, they fall outside the scope of this paper.

**Sequential Quadratic Programming (SQP)**  Originally proposed in [80, 37, 65], SQP serves as a powerful optimization algorithm for nonconvex constrained optimization. Generally speaking, the algorithm talks the following form (cf. [11])

$$x_{t+1} = \arg\min_x \nabla f(x_t)^\top (x - x_t) + \frac{1}{2}(x - x_t)^\top H(x_t)(x - x_t)$$

$$s.t. \quad h(x_t) + J_h(x_t)(x - x_t) = (\leq) 0.$$

For faster convergence, matrix $H(x_t)$ is generally set as the Hessian matrix of the objective function $f$. However, in the setting where only first order information is available, [19] has employed Broyden-Fletcher-Goldfarb-Shanno (BFGS) quasi-Newton Hessian approximations. In our FL-proximal algorithm, the matrix $H(x_t)$ as the scalar matrix $H(x_t) = \frac{1}{\eta}I$, which is also a common choice when second-order information is unavailable [58]. We would also like to note that as far as we know, the acceleration of SQP methods are generally achieved via Newton or quasi-Newton methods, there's little work on exploring acceleration via momentum approaches, which makes our proposed momentum algorithm (18),(19) a novel contribution.

Note that our convergence results rely on the linear independent constraint qualification (LICQ) assumption (Assumption 3). In literature, there are works that seek to replace this restrictive assumption with milder conditions while still maintaining convergence properties, which is generally known as the stabilized SQP (cf. [32, 33, 36, 81, 86]). It is an interesting open question whether we can borrow insights from stabilized SQP to modify our FL algorithms such that the LICQ assumption can be removed.

**Other Constrained Optimization Algorithms** Apart from SQP methods, there are also other optimization algorithms for solving constrained optimization. Interior point method (IPM, cf. [21, 64]) is one of the most deployed algorithm for nonconvex constrained optimization. The algorithm involves navigating through the interior of the feasible region to reach an optimal solution. A common approach within IPMs involves the use of barrier functions to prevent iterates from approaching the boundary of the feasible region. By incorporating these barrier terms into the objective function, the algorithm ensures that each iteration remains within the feasible region's interior. Newton's method is then applied to solve the modified optimization problem, iteratively updating the solution estimate until convergence criteria are met. This methodology allows IPMs to efficiently handle large-scale optimization problems with numerous constraints. While both IPMs and SQP methods can address similar optimization challenges, they differ fundamentally in their approaches: IPMs focus on maintaining iterates within the interior of the feasible region using barrier functions, whereas SQP methods iteratively solve quadratic approximations, often employing active-set strategies to handle constraints.

Augmented Lagrangian Methods (ALMs) (cf. [10, 18]), also known as the method of multipliers, is another popular approach to solve large-scale constrained optimization problems by transforming them into a series of unconstrained problems. This transformation is achieved by augmenting the standard Lagrangian function with a penalty term that penalizes constraint violations, thereby facilitating convergence to the optimal solution.

Further, in convex optimization, proximal gradient methods (cf. [13, 60]) can be applied for a wide class of optimization problems that take the form of

$$\min_x f(x) + g(x)$$

where $f(x)$ is a smooth convex function and $g(x)$ is a convex but potentially nonsmooth, non-differentiable function. The proximal gradient method iteratively updates the solution estimate by:

$$x_{t+1} = \text{prox}_{\eta g}(x_t - \nabla f(x_t)),$$

where $\eta$ is the stepsize and $\text{prox}_{\eta g}$ denotes the proximal operator associated with $g$ defined as:

$$\text{prox}_{\alpha g}(x) = \arg\min_y \left\{ g(y) + \frac{1}{2\alpha} \|y - x\|^2 \right\}$$

In particular, for convex constrained optimization, we can set $g(x)$ as the indicator function:

$$g(x) := \begin{cases} 0 & \text{if } h(x) = (\leq) 0 \\ +\infty & \text{Otherwise} \end{cases}$$

and in this setting, the proximal operator is equivalent to projection onto the feasible set $h(x) = (\leq) 0$. One limitation of the proximal gradient type of algorithms is that it is hard to generalize to the nonconvex setting and that the projection onto the feasible set might be hard to compute the proximal operator for complex $h(x)$.

**Acceleration Algorithms** First-order acceleration methods, such as momentum and Nesterov Accelerated Gradient (NAG), are key advancements in gradient-based optimization. Momentum, introduced by [63], enhances gradient descent by incorporating an extrapolation step that accelerates convergence. Specifically, the update rule $w_t = (1 + \beta)x_t - \beta x_{t-1}$ extrapolates the current position $x_t$ by considering the previous position $x_{t-1}$, effectively predicting a future point in the parameter space. This anticipatory step allows the optimization process to gain speed, particularly in regions where the objective function exhibits gentle slopes or low curvature. Nesterov's Accelerated Gradient [56] refines momentum by proposing a more refined extrapolation parameter $\beta_t$ that possibly varies with iterations, enabling more precise adjustments and achieving faster convergence rates.

Note that momentum or Nesterov acceleration can also be applied to proximal gradient methods (cf. [9, 71]), i.e.

$$y_t = x_t + \beta_t(x_t - x_{t-1})$$
$$x_{t+1} = \text{prox}_{\alpha g}(y_t - \eta \nabla f(y_t)).$$

By integrating a momentum term, it is able to accelerate convergence while maintaining the benefits of the proximal operator in handling nonsmooth $g(x)$. We would also like to note that when $g(x)$ is chosen as the indicator function of $h(x) = Ax + b = 0$ then the momentum accelerated proximal gradient method recovers our momentum SQP scheme (18).

# B  More Numerical Simulations

## B.1  Comparison with other first-order methods

Here we also compare the performance of our algorithm with existing first order methods such as primal dual gradient descent (PDGD) and Augmented Lagrangian method (ALM), as show in the figure below:

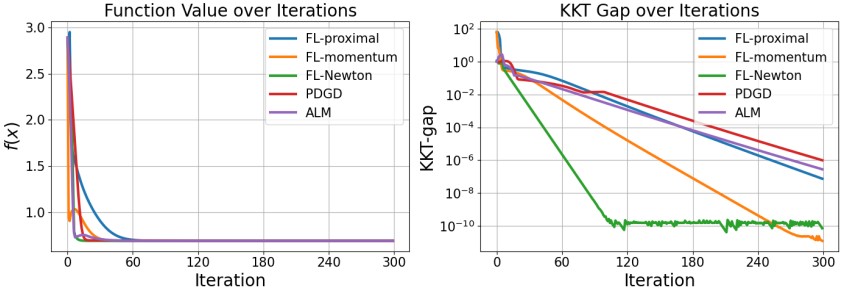

Figure 3: Running different constrained optimization algorithm for Problem (22)

From the above figure, we note that our FL-proximal method performs comparably with existing methods such as PDGD and ALM. And as expected, FL-momentum converges faster as it leverages momentum acceleration and FL-Newton performs the best as it leverages the second order information.

## B.2  Generalization of the FL algorithm using PI Control

As pointed out by Remark 1, one drawback of the FL-proximal is that it requires calculation of the inverse matrix of $J_h(x)J_h(x)^\top$, which might be inefficient when the number of constraints becomes larger. In reality, there are ways to solve the inverse of $J_h(x)J_h(x)^\top$ approximately using heuristic methods or conjugate gradient.

Assume that instead of calculating $(J_h(x)J_h(x)^\top)^{-1}$ we approximate it with some matrix $F(x) \approx (J_h(x)J_h(x)^\top)^{-1}$. For example, in the setting where $J_h(x)J_h(x)^\top$ is diagonal dominant, we can heuristically set $F(x)$ to be a diagonal matrix where the diagonal element of $F(x)$ is the inverse of the corresponding entry of $J_h(x)J_h(x)^\top$, i.e.,

$$F(x) := \left(\text{diag}(J_h(x)J_h(x)^\top)\right)^{-1}. \tag{23}$$

Thus, it is of great importance to study the algorithm's robustness towards the approximation error of $F(x)$. We are going to see in this section that it might be beneficial to consider a PI controller for feedback linearization.

We begin by introducing the Approximated FL algorithm for equality constrained optimization as follows:

---
**Approximated FL**

$$\dot{x} = -\left(\nabla f(x) + J_h(x)^\top \lambda\right)$$
$$\lambda = -F(x)(J_h(x)T(x)\nabla f(x) - Kh(x))$$

---

Note that the only difference of Approximated FL comparing with FL-proximal is that we replace $(J_h(x)J_h(x)^\top)^{-1}$ with its approximation $F(x)$.

As mentioned in Remark 2, apart from considering the feedback linearization with a proportional controller, i.e. $\dot{y} = -Ky$, we can also consider a more general algorithm variant where we also include the integral term and thus formulate the dynamics as

$$\dot{y} = -K_p y - K_i \int_0^t y(s)ds$$

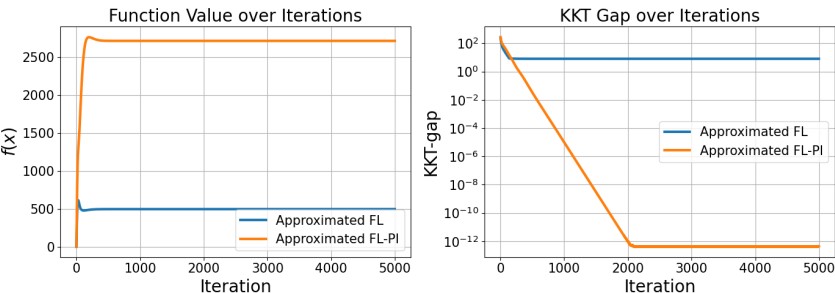

Figure 4: Comparison of Approximated FL and Approximated FL-PI algorithm

This gives rises to the (Approximated) FL-PI algorithm as follows:

**Approximated FL-PI**

$$\dot{x} = -\left(\nabla f(x) + J_h(x)^\top \lambda\right)$$

$$\lambda = -F(x)\left(J_h(x)T(x)\nabla f(x) - \left(K_p h(x) + K_i \int_{s=0}^{t} h(x(s))ds\right)\right)$$

To test the robustness of the algorithm towards approximation error, we run both Approximated FL and Approximated FL-PI algorithm on a quadratic programming problem

$$\min_x \frac{1}{2}x^\top Q x + c^\top x$$
$$s.t. \ Ax + b \leq 0$$

where $m = 10, n = 20$, and $A, b, c, Q^{1/2}$ are all random matrices or vectors whose entries are generated following an i.i.d random Gaussian distribution.

Figure 4 demonstrates the algorithm performance for both Approximated FL and Approximated FL-PI. Interestingly, in this setting, Approximated FL fails to converge to the optimal solution, whereas Approximated FL-PI is able to find the optimal solution. This indicates that it is beneficial to include an additional integral controller term into the optimization algorithm.

### B.3 More numerical examples: Optimal Power Flow

The Alternating Current Optimal Power Flow (AC OPF) problem is a fundamental optimization task in power systems. Its goal is to determine the most efficient operating conditions while satisfying system constraints. This involves optimizing the generation and distribution of electrical power to minimize costs, losses, or other objectives while ensuring that physical laws (such as power flow equations) and operational limits are respected, thus it can be summarized as the following constrained optimization problem:

$$\min_x f(x), \ s.t. \ h_{\text{eq}}(x) = 0, \ h_{\text{ineq}}(x) \leq 0, \tag{24}$$

where the objective function $f(x)$ represents the power generation cost and the equality constraints $h_{\text{eq}}(x)$ generally represents the physical law of the power system, i.e., the power flow equations and $h_{\text{ineq}}$ includes operational limits in terms of voltage, power generation, transmission capacities etc. The optimization variable $x$ generally consists of voltage angles and magnitudes at each bus, and the real and reactive power injections at each generator (see [50] for a detailed introduction on AC OPF).

We solve the AC OPF problem (24) by running the FL-proximal algorithm, FL-Newton algorithm, and FL-momentum algorithm. Figure 5 presents the numerical results for solving AC OPF on the IEEE-39 and IEEE-118 bus systems, respectively. In both cases, FL-Newton demonstrates the fastest convergence, which is expected given that it leverages second-order information (i.e., the Hessian). Comparing FL-proximal and FL-momentum, both of which rely solely on first-order information, Figure 5 indicates that FL-momentum accelerates the learning process and achieves faster convergence than FL-proximal for the IEEE-39 bus system. However, in the IEEE-118 bus

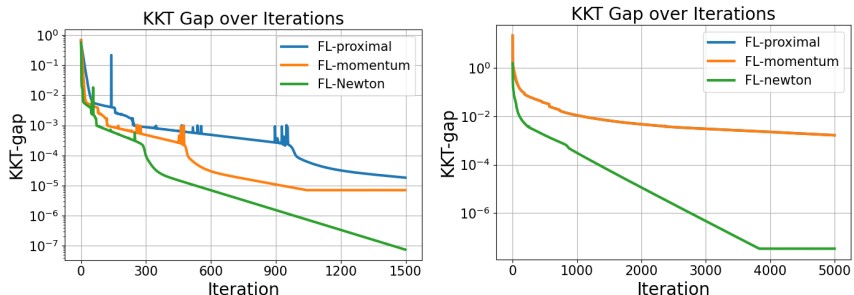

Figure 5: Result for AC OPF on
IEEE-39 bus (left) and IEEE-118 bus (right) bus system

system, FL-proximal and FL-momentum exhibit similar convergence speeds, with their learning curves nearly overlapping. We hypothesize that this problem is ill-conditioned, limiting the effectiveness of momentum in accelerating the algorithm.

## C Proof of Theorem 1

*Proof of Theorem 1.* Statement 1 is obvious from the derivation in Section 2 because we have that $\dot{y}_i = -k_i y_i$, thus $y_i(t) = e^{-k_i t} y_i(0)$ always keeps the same sign, since $y_i(t) = h_i(x(t))$, we have completed the proof for statement 1.

For statement 2, we first derive the time derivative for $f(x)$. For notational simplicity we define $P(x) := T(x) - T(x) J_h(x)^\top \left( J_h(x) T(x) J_h(x)^\top \right)^{-1} J_h(x) T(x)$. Note that $P(x) \succeq 0$ and that $P(x) T(x)^{-1} P(x) = P(x)$.

$$
\begin{aligned}
\dot{f}(x) &= \nabla f(x)^\top \dot{x} \\
&= -\nabla f(x)^\top T(x) \nabla f(x) - \nabla f(x)^\top T(x) J_h(x)^\top \lambda \\
&= -\nabla f(x)^\top T(x) \nabla f(x) + \nabla f(x)^\top T(x) J_h(x)^\top \left( J_h(x) T(x) J_h(x)^\top \right)^{-1} \left( J_h(x) T(x) \nabla f(x) - K h(x) \right) \\
&= -\nabla f(x)^\top P(x) \nabla f(x) - \nabla f(x)^\top T(x) J_h(x)^\top \left( J_h(x) T(x) J_h(x)^\top \right)^{-1} K h(x) \\
&\le -\nabla f(x)^\top P(x) \nabla f(x) + \| \nabla f(x)^\top T(x) J_h(x)^\top \left( J_h(x) T(x) J_h(x)^\top \right)^{-1} \|_\infty \sum_{i=1}^n k_i |h_i(x)| \\
&\le -\nabla f(x)^\top P(x) \nabla f(x) + \frac{\lambda_{\max}}{\lambda_{\min}} (MD)^2 \sum_{i=1}^n k_i |h_i(x)|.
\end{aligned}
$$

Further, from statement 1 we have that

$$
\frac{d|h_i(x)|}{dt} = -k_i |h_i(x)|,
$$

thus we have

$$
\begin{aligned}
&\frac{d \left( f(x) + \frac{\lambda_{\max}}{\lambda_{\min}} MD \sum_{i=1}^m |h_i(x)| \right)}{dt} \\
&\le -\nabla f(x)^\top P(x) \nabla f(x) + \frac{\lambda_{\max}}{\lambda_{\min}} (MD)^2 \sum_{i=1}^n k_i |h_i(x)| - \frac{\lambda_{\max}}{\lambda_{\min}} (MD)^2 \sum_{i=1}^n k_i |h_i(x)| \\
&\le -\nabla f(x)^\top P(x) \nabla f(x) \le 0,
\end{aligned}
\tag{25}
$$

i.e., $\ell(x(t)) := f(x(t)) + (MD)^2 \sum_{i=1}^m |h_i(x(t))|$ is non-increasing, which completes the proof. Note that $\ell(x(t))$ is always differentiable with respect to $t$ because based on Statement 1, $x(t)$ will always stay in $\mathcal{E}_i$ (or $\mathcal{E}_i^c$) if the initialization $x(0) \in \mathcal{E}_i$ (or $\mathcal{E}_i^c$).

Now we prove statement 3. Since $f$ is lower bounded and that $h(x) \to 0$ asymptotically, we arrive at the conclusion that the trajectory of $\ell(x(t))$ is bounded. From (25) we have that

$$
\frac{d\ell(x(t))}{dt} \le -\nabla f(x(t))^\top P(x(t)) \nabla f(x(t))
$$

$$\implies \ell(x(T)) - \ell(x(0)) \leq -\int_{t=0}^{T} \nabla f(x(t))^{\top} P(x(t)) \nabla f(x(t)) dt = -\int_{t=0}^{T} \|P(x(t)) \nabla f(x(t))\|_{T(x(t))^{-1}}^2 dt$$

$$\implies \int_{t=0}^{T} \|P(x(t)) \nabla f(x(t))\|_{T(x(t))^{-1}}^2 dt \leq (\ell(x(0)) - \ell(x(T))).$$

Further, it is not hard to verify that

$$P(x(t)) \nabla f(x(t)) = T(x(t)) \nabla f(x(t)) - T(x) J_h(x)^{\top} \left( J_h(x) T(x) J_h(x)^{\top} \right)^{-1} J_h(x) T(x) \nabla f(x(t))$$
$$= T(x(t)) \left( \nabla f(x(t)) + J_h(x(t))^{\top} \overline{\lambda}(t) \right),$$

so we have

$$\int_{t=0}^{T} \|P(x(t)) \nabla f(x(t))\|_{T(x(t))^{-1}}^2 dt = \int_{t=0}^{T} \|\nabla f(x(t)) + J_h(x(t))^{\top} \overline{\lambda}(t)\|_{T(x(t))}^2 dt \leq \ell(x(0)) - \ell(x(T)).$$

$$\implies \int_{t=0}^{T} \|\nabla f(x(t)) + J_h(x(t))^{\top} \overline{\lambda}(t)\|^2 dt \leq \frac{1}{\lambda_{\min}} (\ell(x(0)) - \ell(x(T)))$$

We also have that

$$\|\lambda(t) - \overline{\lambda}(t)\| = \| \left( J_h(x) T(x) J_h(x)^{\top} \right)^{-1} K h(x(t))\| \leq \frac{1}{\lambda_{\min}} D^2 \|K\| \|h(x_t)\| \to 0, \text{ as } t \to +\infty$$

which completes the proof of statement 3.

We now prove statement 4. From statement 3 we have that

$$\inf_{\frac{T}{2} \leq t \leq T} \|\nabla f(x(t)) + J_h(x(t))^{\top} \overline{\lambda}(t)\|^2 \leq \frac{2}{\lambda_{\min} T} \ell(x(0)) - \ell(x(T))$$

$$\leq \frac{2}{T} \left( \frac{f(x(0)) - f_{\min}}{\lambda_{\min}} + \frac{\lambda_{\max} M^2 D^2}{\lambda_{\min}^2} \sum_i |h_i(x(0))| \right)$$

$$\implies \inf_{\frac{T}{2} \leq t \leq T} \|\nabla f(x(t)) + J_h(x(t))^{\top} \overline{\lambda}(t)\| \leq \sqrt{\frac{2}{T} \left( \frac{f(x(0)) - f_{\min}}{\lambda_{\min}} + \frac{\lambda_{\max} M^2 D^2}{\lambda_{\min}^2} \sum_i |h_i(x(0))| \right)}.$$

Further, from statement 1

$$|h_i(x(t))| \leq h_i(x(0)) e^{-\frac{k_i T}{2}}, \quad \text{for all } t \geq \frac{T}{2},$$

thus

$$\inf_{\frac{T}{2} \leq t \leq T} \texttt{KKT-gap}(x(t), \overline{\lambda}(t)) = \inf_{\frac{T}{2} \leq t \leq T} \max\{\|\nabla f(x(t)) + J_h(x(t))^{\top} \overline{\lambda}(t)\|, \max_{1 \leq i \leq m} \{\|h(x(t))\|_{\infty}\}\}$$

$$\leq \max \left\{ \sqrt{\frac{2}{T} \left( \frac{f(x(0)) - f_{\min}}{\lambda_{\min}} + \frac{\lambda_{\max} M^2 D^2}{\lambda_{\min}^2} \sum_i |h_i(x(0))| \right)}, \max_{1 \leq i \leq m} \left\{ h_i(x(0)) e^{-\frac{k_i T}{2}} \right\} \right\}.$$

Additionally, it is not hard to verify from statement 3 and statement 1 that

$$\lim_{t \to +\infty} \|\nabla f(x(t)) + J_h(x(t))^{\top} \overline{\lambda}(t)\| = 0, \lim_{t \to +\infty} \|h(x(t))\|_{\infty} = 0, \lim_{t \to +\infty} \|\overline{\lambda}(t) - \lambda(t)\| = 0$$

thus

$$\lim_{t \to +\infty} \texttt{KKT-gap}(x(t), \overline{\lambda}(t)) = 0, \quad \lim_{t \to +\infty} \texttt{KKT-gap}(x(t), \lambda(t)) = 0.$$

$\square$

## C.1  Global Convergence for Strongly Convex Settings

Under the assumption that the optimization problem (1) is strongly convex, we are able to establish the result that FL-proximal converges to the global optimal solution with a linear rate. We make the following assumption.

**Assumption 7** (Strong Convexity). *The objective function $f(x)$ is strongly convex, i.e. for any $x, y \in \mathbb{R}^n$,*

$$f(x) - f(y) \geq \nabla f(y)^\top (x - y) + \frac{\mu}{2}\|x - y\|^2.$$

*The constraint function $h(x)$ in (1) takes the form of $h(x) = Ax + b$ where $A \in \mathbb{R}^{m \times n}, b \in \mathbb{R}^m$, and $A$ is of full row rank.*

**Theorem 6.** *Under Assumption 7, we have that running FL-proximal ((6) with $T(x) = I$) will give*

$$f(x) - f(x^\star) \leq e^{-2\mu t}(f(x(0)) - f(x^\star)),$$

*where $x^\star$ is the global optimal solution of (1).*

*Proof.* We denote $P_A := A^\top(AA^\top)^{-1}A, P_A^\perp = I - P_A$. Then the key argument is that according to strong convexity we have:

$$\begin{aligned}
f(x) - f(x^\star) &\leq \nabla f(x)^\top (x - x^\star) - \frac{\mu}{2}\|x - x^\star\|^2 \\
&\leq \nabla f(x)^\top P_A^\perp (x - x^\star) - \frac{\mu}{2}\|P_A^\perp(x - x^\star)\|^2 + \nabla f(x)^\top P_A(x - x^\star) \\
&\leq \frac{1}{2\mu}\nabla f(x)P_A^\perp \nabla f(x) + \nabla f(x)P_A(x - x^\star).
\end{aligned}$$

Further,

$$\begin{aligned}
\frac{df}{dt} &= -\nabla f(x)^\top(\nabla f(x) + J_h(x)^\top \lambda) \\
&= -\nabla f(x)^\top(\nabla f(x) - A^\top(AA^\top)^{-1}(A\nabla f(x) - (Ax + b))) \qquad (J_h(x) = A) \\
&= -\nabla f(x)^\top P_A^\perp \nabla f(x) - \nabla f(x)^\top A^\top(AA^\top)^{-1}A^\top(Ax + b) \\
&= -\nabla f(x)^\top P_A^\perp \nabla f(x) - \nabla f(x)^\top A^\top(AA^\top)^{-1}A^\top(Ax - Ax^\star) \\
&= -(\nabla f(x)P_A^\perp \nabla f(x) + \nabla f(x)P_A(x - x^\star)),
\end{aligned}$$

and thus we have

$$\frac{d(f(x(t)) - f(x^\star))}{dt} \leq -2\mu(f(x(t)) - f(x^\star)),$$

which completes the proof $\qquad\qquad\square$

## D   Proof of Theorem 2 and 3

*Proof of Theorem 2.* We write out the lagrangian for the SQP problem (9)

$$L(x, \lambda) = \nabla f(x_t)^\top (x - x_t) + \frac{1}{2\eta}(x - x_t)^\top T(x_t)^{-1}(x - x_t) + \lambda^\top (h(x_t) + J_h(x_t)(x - x_t)).$$

Since the optimization problem (9) is convex and satisfies the relaxed Slater condition, the KKT condition is necessary and sufficient for global optimality. The KKT condition implies that

$$\begin{aligned}
\partial_x L(x, \lambda) = 0 &\implies \nabla f(x_t) + \frac{1}{\eta}T(x_t)^{-1}(x - x_t) + J_h(x_t)^\top \lambda = 0 \\
&\implies x - x_t = -T(x_t)\eta(\nabla f(x_t) + J_h(x_t)^\top \lambda).
\end{aligned}$$

Since $h(x_t) + J_h(x_t)(x - x_t) = 0$, we have

$$h(x_t) + J_h(x_t)(x - x_t) = h(x_t) - \eta J_h(x_t)T(x_t)(\nabla f(x_t) + J_h(x_t)^\top \lambda) = 0$$

$$\implies \left(J_h(x_t)T(x_t)J_h(x_t)^\top\right)\lambda = \frac{1}{\eta}h(x_t) - J_h(x_t)T(x_t)\nabla f(x_t)$$

$$\implies \lambda = -\left(J_h(x_t)T(x_t)J_h(x_t)^\top\right)^{-1}\left(J_h(x_t)T(x_t)\nabla f(x_t) - \frac{1}{\eta}h(x_t)\right).$$

Thus we have that

$$x_{t+1} = x_t - \eta T(x_t)\left(\nabla f(x_t) - J_h(x_t)^\top\left(J_h(x_t)T(x_t)J_h(x_t)^\top\right)^{-1}\left(J_h(x_t)T(x_t)\nabla f(x_t) - \frac{1}{\eta}h(x_t)\right)\right).$$

This is exatcly (8) with $K = \frac{1}{\eta}$. $\qquad\qquad\square$

*Proof of Theorem 3.* We write out the Lagrangian for the convex optimization problem (15)

$$L(x, \lambda) = \nabla f(x_t)^\top (x - x_t) + \frac{1}{2\eta}(x-x_t)^\top T(x_t)^{-1}(x-x_t)$$
$$+ \lambda(h(x_t) + J_h(x_t)(x - x_t)).$$

Since (15) is feasible, it satisfies the relaxed Slater condition, thus we have that strong duality holds for (15). Also note that the optimization problem in (14)

$$\lambda^\star := \arg\min_{\lambda \geq 0} \left( \frac{1}{2} \lambda^\top J_h(x_t)T(x_t)J_h(x_t)^\top \lambda \right.$$
$$\left. + \lambda^\top \left( J_h(x_t)T(x_t)\nabla f(x_t) - Kh(x_t) \right) \right)$$

is the dual problem of (15) (see Lemma 2 in Appendix G). Thus the solution of $\lambda^\star$ in (14) serves as the Lagrangian dual variable of (9). Then, we have that the $x^\star$ is an optimal solution of (15) if and only if

$$x^\star = \arg\min_x L(x, \lambda^\star) \iff \partial_x L(x, \lambda^\star) = 0$$
$$\iff \nabla f(x_t) + \frac{1}{\eta} T(x_t)^{-1}(x^\star - x_t) + J_h(x_t)^\top \lambda^\star = 0$$
$$\iff x^\star - x_t = -\eta T(x_t)(\nabla f(x_t) + J_h(x_t)^\top \lambda^\star),$$

thus we have that both (14) and (15) follow the update:

$$x_{t+1} - x_t = -\eta T(x_t)(\nabla f(x_t) + J_h(x_t)^\top \lambda^\star),$$

which completes the proof. $\square$

# E   Proof of Theorem 4

Before proving Theorem 4, we first introduce the following lemma which plays an important role in the proof.

**Lemma 1.** *Solving $\lambda$ in* (13) *is equivalent to solving the following equations:*

$$J_h(x)T(x)J_h(x)^\top \lambda + J_h(x)T(x)\nabla f(x) = Ky + s,$$
$$s^\top \lambda = 0 \tag{26}$$
$$s \geq 0, \ \lambda \geq 0$$

*Proof.* The proof is simply writing out the KKT condition of the optimization problem. $\square$

We are now ready to prove Theorem 4

*Proof of Theorem 4.* We first prove statement 1.

$$\dot{y} = J_h(x)\dot{x} = -J_h(x)T(x)\nabla f(x) - J_h(x)T(x)J_h(x)^\top \lambda$$
$$= -Ky - s \ \text{(Lemma 1)}$$

thus

$$\dot{y}_i = -k_i y_i - s \leq -k_i y_i$$

and hence we have proven statement 1.

Now we prove statement 2.

$$\dot{f}(x) = \nabla f(x)^\top \dot{x} = -\nabla f(x)^\top T(x)\nabla f(x) - \lambda^\top J_h(x)T(x)\nabla f(x)$$
$$= -\nabla f(x)^\top T(x)\nabla f(x) - \lambda^\top J_h(x)T(x)\nabla f(x) - \lambda^\top Ky + \lambda^\top Ky$$
$$= -\nabla f(x)^\top T(x)\nabla f(x) - \lambda^\top J_h(x)T(x)\nabla f(x) - \lambda^\top J_h(x)T(x)J_h(x)^\top \lambda - \lambda^\top J_h(x)T(x)\nabla f(x) + \lambda^\top Ky$$

$$= -\|\nabla f(x) + J_h(x)^\top \lambda\|^2_{T(x)} + \lambda^\top K y$$

$$= -\|\nabla f(x) + J_h(x)^\top \lambda\|^2_{T(x)} + \lambda_{\mathcal{I}^c}^\top (Ky)_{\mathcal{I}^c} + \lambda_{\mathcal{I}}^\top (Ky)_{\mathcal{I}}$$

$$\leq -\|\nabla f(x) + J_h(x)^\top \lambda\|^2_{T(x)} + \lambda_{\mathcal{I}^c}^\top (Ky)_{\mathcal{I}^c} + \|\lambda\|_\infty \sum_{i \in \mathcal{I}} k_i y_i$$

$$\leq -\|\nabla f(x) + J_h(x)^\top \lambda\|^2_{T(x)} + \lambda_{\mathcal{I}^c}^\top (Ky)_{\mathcal{I}^c} + L \sum_{i \in \mathcal{I}} k_i y_i$$

Here we abbreviate $\mathcal{I}(x) = \{i | h_i(x) > 0, 1 \leq i \leq m\}$ as $\mathcal{I}$. From the proof of Statement 1 we have that for $i \in \mathcal{I}$

$$\dot{h}_i(x) = -k_i y_i - s_i \leq -k_i y_i$$

Thus, combining the inequalities together, we have that

$$\frac{d\left(f(x) + L\sum_{i \in \mathcal{I}} h_i(x)\right)}{dt} \leq -\|\nabla f(x) + J_h(x)^\top \lambda\|^2_{T(x)} + \lambda_{\mathcal{I}^c}^\top (Ky)_{\mathcal{I}^c} + L \sum_{i \in \mathcal{I}} k_i y_i - L \sum_{i \in \mathcal{I}} k_i y_i$$

$$\leq -\|\nabla f(x) + J_h(x)^\top \lambda\|^2_{T(x)} + \lambda_{\mathcal{I}^c}^\top (Ky)_{\mathcal{I}^c} \leq 0 \quad (y_i \leq 0 \text{ for } i \in \mathcal{I}^c). \tag{27}$$

Further, we have that the function on the righthand side is equivalent to

$$f(x) + L \sum_{i \in \mathcal{I}} h_i(x) = f(x) + L \sum_i [h_i(x)]_+ = \ell(x).$$

We would also like to note that given $f(x)$ and $h(x)$ are differentiable and Lipschitz, we have that $\ell(x)$ is an absolute continuous function (cf. [69]) and almost everywhere differentiable. Hence, given that $\frac{d\ell(x(t))}{dt} \leq 0$ holds almost everywhere, we have that $\ell(x(t))$ is non-increasing with respect to $t$ which completes the proof.

We now prove Statement 3. From (27) we have that

$$\dot{\ell}(x) \leq -\|\nabla f(x) + J_h(x)^\top \lambda\|^2_{T(x)} + \lambda_{\mathcal{I}^c}^\top (Ky)_{\mathcal{I}^c}$$

$$\implies \int_{t=0}^T \left( \|\nabla f(x(t)) + J_h(x(t))\lambda(t)\|^2_{T(x)} - \sum_{i \in \mathcal{I}(x)^c} k_i \lambda_i(t) h_i(x(t)) \right) dt \leq \ell(x(0)) - \ell(x(T)),$$

which completes the proof. Note that here we leverage the fact that $\ell(x)$ is absolute continuous.

We now prove Statement 4. From Statement 3 we have that

$$\inf_{\frac{T}{2} \leq t \leq T} \left( \|\nabla f(x(t)) + J_h(x(t))\lambda(t)\|^2_{T(x)} - \sum_{i \in \mathcal{I}(x)^c} k_i \lambda_i(t) h_i(x(t)) \right) \leq \frac{2}{T} \left( \ell(x(0)) - \ell(x(T)) \right)$$

$$\leq \frac{2}{T} \left( f(x(0)) - f_{\min} + L \sum_{i \in \mathcal{I}(x(0))} h_i(x(0)) \right)$$

Thus we have that there exists a $\frac{T}{2} \leq t^\star \leq T$ such that

$$\|\nabla f(x(t^\star)) + J_h(x(t^\star))\lambda(t^\star)\|^2_{T(x)} - \sum_{i \in \mathcal{I}(x(t^\star))^c} k_i \lambda_i(t^\star) h_i(x(t^\star)) \leq \frac{2}{T} \left( f(x(0)) - f_{\min} + L \sum_{i \in \mathcal{I}(x(0))} h_i(x(0)) \right)$$

Thus

$$\|\nabla f(x(t^\star)) + J_h(x(t^\star))\lambda(t^\star)\| \leq \sqrt{\frac{2}{\lambda_{\min} T} \left( f(x(0)) - f_{\min} + L \sum_{i \in \mathcal{I}(x(0))} h_i(x(0)) \right)},$$

$$-\sum_{i \in \mathcal{I}(x(t^\star))^c} k_i \lambda_i(t^\star) h_i(x(t^\star)) \leq \frac{2}{T} \left( f(x(0)) - f_{\min} + L \sum_{i \in \mathcal{I}(x(0))} h_i(x(0)) \right).$$

Since $t^\star \geq \frac{T}{2}$ we have that

$$h_i(x(t^\star)) \leq h_i(x(0))e^{-\frac{k_i T}{2}},$$
$$\sum_{i \in \mathcal{I}(x(t^\star))} \lambda_i(t^\star)h_i(x(t^\star)) \leq Le^{-\frac{\min_i k_i T}{2}} \sum_{i \in \mathcal{I}(x(0))} h_i(x(0)).$$

Thus we have that

$$|\lambda(t^\star)^\top h(x(t^\star))| = -\sum_{i \in \mathcal{I}(x(t^\star))^c} \lambda_i(t^\star)h_i(x(t^\star)) + \sum_{i \in \mathcal{I}(x(t^\star))} \lambda_i(t^\star)h_i(x(t^\star))$$

$$\leq \frac{1}{\min_i k_i} \frac{2}{T}\left(f(x(0)) - f_{\min} + L\sum_{i \in \mathcal{I}(x(0))} h_i(x(0))\right) + Le^{-\frac{\min_i k_i T}{2}} \sum_{i \in \mathcal{I}(x(0))} h_i(x(0))$$

$$\leq \frac{1}{\min_i k_i} \frac{2}{T}\left(f(x(0)) - f_{\min} + (L+1)\sum_{i \in \mathcal{I}(x(0))} h_i(x(0))\right)$$

Thus we have that

$$\inf_{0 \leq t \leq T} \texttt{KKT-gap}(x(t), \lambda(t)) \leq \texttt{KKT-gap}(x(t^\star), \lambda(t^\star))$$

$$= \max\left\{\|\nabla f(x(t^\star)) + J_h(x(t^\star))^\top \lambda(t^\star)\|, \left|\lambda(t^\star)^\top h(x(t^\star))\right|, \max_i[h_i(x(t^\star))]_+\right\}$$

$$\leq \max\left\{\sqrt{\frac{2}{\lambda_{\min}T}\left(f(x(0)) - f_{\min} + L\sum_{i \in \mathcal{I}(x(0))} h_i(x(0))\right)},\right.$$

$$\left. \frac{1}{\min_i k_i} \frac{2}{T}\left(f(x(0)) - f_{\min} + (L+1)\sum_{i \in \mathcal{I}(x(0))} h_i(x(0))\right)\right\}$$

Further, from statement 3 we get that

$$\lim_{t \to +\infty} \|\nabla f(x(t)) + J_h(x(t))\lambda(t)\| = 0,$$
$$\lim_{t \to +\infty} -\sum_{i \in \mathcal{I}(x)^c} k_i\lambda_i(t)h_i(x(t)) = 0$$

From statement 1 we get that

$$\lim_{t \to +\infty} [h_i(x(t))]_+ = 0$$
$$\lim_{t \to +\infty} \sum_{i \in \mathcal{I}(x)} \lambda_i(t)h_i(x(t)) = 0.$$

Additionally,

$$\lim_{t \to +\infty} |\lambda(t)h(x(t))| \leq -\frac{1}{\min_i k_i} \lim_{t \to +\infty} \sum_{i \in \mathcal{I}(x)^c} k_i\lambda_i(t)h_i(x(t)) + \lim_{t \to +\infty} \sum_{i \in \mathcal{I}(x)} \lambda_i(t)h_i(x(t)) = 0,$$

thus we have that

$$\lim_{t \to +\infty} \texttt{KKT-gap}(x(t), \lambda(t)) = 0,$$

which completes the proof. $\qquad\square$

# F   Proof of Theorem 5

We first define the following notations: denote $P(x) := I - J_h(x)^\top (J_h(x)J_h(x)^\top)^{-1} J_h(x)$. Note that we have $P(x) = P(x)^2$.

*Proof of Theorem 5.* From (19) we have that

$$\frac{df(x)}{dt} = \nabla f(x)^\top z \tag{28}$$

$$\begin{aligned}
\frac{d^2 f(x)}{dt^2} &= \frac{d(\nabla f(x)^\top z)}{dt}\\
&= z^\top \nabla^2 f(x) z + \nabla f(x)^\top (-\alpha z - \nabla f(x) - J_h(x)^\top \lambda)\\
&= z^\top \nabla^2 f(x) z - \alpha \nabla f(x)^\top z - \nabla f(x)^\top P(x) f(x) - \nabla f(x)^\top J_h(x)^\top (J_h(x) J_h(x)^\top)^{-1} K h(x)\\
&= z^\top \nabla^2 f(x) z - \alpha \nabla f(x)^\top z - \nabla f(x)^\top P(x) f(x) + \bar{\lambda}(x)^\top K h(x) \tag{29}
\end{aligned}$$

Thus by combining $\alpha \times (28) + (29)$ we have

$$\frac{d}{dt}\left(\alpha f(x) + \nabla f(x)^\top z\right) = z^\top \nabla^2 f(x) z - \nabla f(x)^\top P(x) f(x) + \bar{\lambda}(x)^\top K h(x) \tag{30}$$

Similarly we have

$$\frac{d\frac{1}{2}\|h(x)\|^2}{dt} = h(x)^\top J_h(x) z \tag{31}$$

$$\begin{aligned}
\frac{d^2 \frac{1}{2}\|h(x)\|^2}{dt^2} &= \frac{h(x)^\top J_h(x) z}{dt}\\
&= z^\top J_h(x)^\top J_h(x) z + h(x)^\top J_h(x)(-\alpha z - \nabla f(x) - J_h(x)^\top \lambda) + \sum_{i=1}^m z_i h(x)^\top \nabla^2 h_i(x) z\\
&= z^\top J_h(x)^\top J_h(x) z - \alpha h(x)^\top J_h(x) z - h(x)^\top J_h(x)(\nabla f(x) + J_h(x)^\top \lambda) + \sum_{i=1}^m z_i h(x)^\top \nabla^2 h_i(x) z \tag{32}
\end{aligned}$$

By the definition of $\lambda$ we know that

$$J_h(x)(\nabla f(x) + J_h(x)^\top \lambda) = K h(x),$$

thus we have

$$\frac{h(x)^\top J_h(x) z}{dt} = z^\top J_h(x)^\top J_h(x) z - \alpha h(x)^\top J_h(x) z - h(x)^\top K h(x) + \sum_{i=1}^m z_i h(x)^\top \nabla^2 h_i(x) z \tag{33}$$

Let $\bar{H}(x) := [h(x)^\top \nabla^2 h_i(x)]_{i=1}^n$, then $\sum_{i=1}^m z_i h(x)^\top \nabla^2 h_i(x) z = z^\top \bar{H}(x) z$. Thus by combining $\alpha \times (31) + (33)$ we have

$$\frac{d}{dt}\left(\frac{\alpha}{2}\|h(x)\|^2 + h(x)^\top J_h(x) z\right) = z^\top J_h(x)^\top J_h(x) z - h(x)^\top K h(x) + z^\top \bar{H}(x) z \tag{34}$$

Another set of ODE we will use is

$$\frac{d\bar{\lambda}(x)^\top h(x)}{dt} = \bar{\lambda}(x)^\top J_h(x) z + h(x)^\top \frac{\partial \bar{\lambda}(x)}{\partial x} z \tag{35}$$

$$\begin{aligned}
&\frac{d}{dt}\left(\bar{\lambda}(x)^\top J_h(x) z + h(x)^\top \frac{\partial \bar{\lambda}(x)}{\partial x} z\right)\\
&= z^\top \left(\frac{\partial \left(J_h(x)^\top \lambda(x)\right)}{\partial x}\right)^\top z + \bar{\lambda}(x)^\top J_h(x)(-\alpha z - \nabla f(x) - J_h(x)^\top \lambda)\\
&\quad + z^\top \left(\frac{\partial \left(h(x)^\top \frac{\partial \bar{\lambda}(x)}{\partial x}\right)}{\partial x}\right)^\top z + h(x)^\top \frac{\partial \bar{\lambda}(x)}{\partial x}(-\alpha z - \nabla f(x) - J_h(x)^\top \lambda)\\
&= z^\top \left(\frac{\partial \left(J_h(x)^\top \lambda(x) + \left(\frac{\partial \bar{\lambda}(x)}{\partial x}\right)^\top h(x)\right)}{\partial x}\right) z - \alpha \bar{\lambda}(x)^\top J_h(x) z - \bar{\lambda}(x)^\top K h(x)
\end{aligned}$$

$$- \alpha h(x)^\top \frac{\partial \bar{\lambda}(x)}{\partial x} z - h(x)^\top \frac{\partial \bar{\lambda}(x)}{\partial x} P(x) \nabla f(x) - h(x)^\top \frac{\partial \bar{\lambda}(x)}{\partial x} J_h(x)^\top (J_h(x) J_h(x)^\top)^{-1} K h(x) \tag{36}$$

Thus by combining $\alpha \times (35) + (36)$ we get

$$\frac{d}{dt} \left( \alpha \bar{\lambda}(x)^\top h(x) + \bar{\lambda}(x)^\top J_h(x) z + h(x)^\top \frac{\partial \bar{\lambda}(x)}{\partial x} z \right)$$

$$= z^\top \left( \frac{\partial \left( J_h(x)^\top \lambda(x) + \left( \frac{\partial \bar{\lambda}(x)}{\partial x}^\top h(x) \right) \right)}{\partial x} \right) z - \bar{\lambda}(x)^\top K h(x)$$

$$- h(x)^\top \frac{\partial \bar{\lambda}(x)}{\partial x} P(x) \nabla f(x) - h(x)^\top \frac{\partial \bar{\lambda}(x)}{\partial x} J_h(x)^\top (J_h(x) J_h(x)^\top)^{-1} K h(x) \tag{37}$$

Further we also have

$$\frac{d \frac{1}{2} \|z\|^2}{dt^2} = z^\top (-\alpha z - P(x) \nabla f(x) - J_h(x)^\top (J_h(x) J_h(x)^\top)^{-1} K h(x))$$

$$= -\alpha \|z\|^2 - z^\top P(x) \nabla f(x) - z^\top J_h(x)^\top (J_h(x) J_h(x)^\top)^{-1} K h(x) \tag{38}$$

Finally, we combine $a_1 \times (30) + a_1 \times (37) + a_2 (34) + (38)$ we get

$$\frac{d}{dt} \underbrace{\left( a_1 \alpha f(x) + \frac{a_2 \alpha}{2} \|h(x)\|^2 + \left( a_1 \nabla f(x) + a_2 J_h(x)^\top h(x) + a_1 J_h(x)^\top \bar{\lambda}(x) + a_1 \frac{\partial \bar{\lambda}(x)}{\partial x}^\top h(x) \right)^\top z + a_1 \alpha \bar{\lambda}(x)^\top h(x) + \|z\|^2 \right)}_{\ell(x,z)}$$

$$= z^\top \left( a_1 \nabla^2 f(x) + a_1 \left( \frac{\partial \left( J_h(x)^\top \lambda(x) + \left( \frac{\partial \bar{\lambda}(x)}{\partial x}^\top h(x) \right) \right)}{\partial x} \right) + a_2 J_h(x)^\top J_h(x) + a_2 \bar{H}(x) \right) z - \alpha \|z\|^2$$

$$\left. \begin{array}{l} - a_1 h(x)^\top \frac{\partial \bar{\lambda}(x)}{\partial x} P(x) \nabla f(x) - a_1 h(x)^\top \frac{\partial \bar{\lambda}(x)}{\partial x} J_h(x)^\top (J_h(x) J_h(x)^\top)^{-1} K h(x) \\ - z^\top P(x) \nabla f(x) - z^\top J_h(x)^\top (J_h(x) J_h(x)^\top)^{-1} K h(x) \\ - a_1 \nabla f(x)^\top P(x) f(x) - a_2 h(x)^\top K h(x) \end{array} \right\}_{\text{Part I}} . \tag{39}$$

Note that for $a_2 \geq 4 a_1 \times \frac{\lambda_{\max}(K)}{\lambda_{\min}(K)} \left\| \frac{\partial \bar{\lambda}(x)}{\partial x} \right\| \sqrt{\|(J_h(x) J_h(x)^\top)^{-1}\|} + \frac{a_1}{\lambda_{\min}(K)} \left\| \frac{\partial \bar{\lambda}(x)}{\partial x} \right\|^2$, we have that

$$- a_2 h(x)^\top K h(x) - a_1 h(x)^\top \frac{\partial \bar{\lambda}(x)}{\partial x} J_h(x)^\top (J_h(x) J_h(x)^\top)^{-1} K h(x) \leq -\frac{3}{4} a_2 h(x)^\top K h(x)$$

$$- \frac{a_1}{2} \nabla f(x)^\top P(x) f(x) - a_1 h(x)^\top \frac{\partial \bar{\lambda}(x)}{\partial x} P(x) \nabla f(x) \leq \frac{a_1}{2} \left\| \frac{\partial \bar{\lambda}(x)}{\partial x} \right\|^2 \frac{1}{\lambda_{\min}(K)} h(x)^\top K h(x)$$

$$\leq \frac{1}{2} a_2 h(x)^\top K h(x)$$

Thus substituting the above equation into Part I of (39) we get

$$\text{Part I} \leq -\frac{a_2}{4} h(x)^\top K h(x) - \frac{a_1}{2} \nabla f(x)^\top P(x) f(x) - z^\top P(x) \nabla f(x) - z^\top J_h(x)^\top (J_h(x) J_h(x)^\top)^{-1} K h(x)$$

$$\leq \frac{1}{a_1} \|z\|^2 + \frac{2}{a_2} \|K\| \left\| (J_h(x) J_h(x)^\top)^{-1} \right\| \|z\|^2 - \frac{a_2}{8} h(x)^\top K h(x) - \frac{a_1}{4} \|P(x) \nabla f(x)\|^2$$

Then, substitute the above inequality to (39) and by setting

$$a_2 \geq a_1 \times \left( 4 \frac{\lambda_{\max}(K)}{\lambda_{\min}(K)} L_2 D + \frac{L_1^2}{\lambda_{\min}(K)} \right)$$

we get

$$\frac{d\ell(x,z)}{dt} \leq \left( a_1(L_f + L_2) + a_2(M^2 + \bar{H}) + \frac{1}{a_1} + \frac{2(\lambda_{\max}(K)D^2)}{a_2} \right) \|z\|^2 - \alpha\|z\|^2$$
$$- \frac{a_2}{8} h(x)^\top K h(x) - \frac{a_1}{4} \nabla f(x)^\top P(x) f(x)$$

Thus by setting

$$\alpha \geq \left( a_1(L_f + L_2) + a_2(M^2 + \bar{H}) + \frac{1}{a_1} + \frac{2(\lambda_{\max}(K)D^2)}{a_2} \right) + 1$$

we get

$$\frac{d\ell(x,z)}{dt} \leq -\|z\|^2 - \frac{a_2}{8} h(x)^\top K h(x) - \frac{a_1}{4} \|P(x)\nabla f(x)\|^2 \leq 0, \tag{40}$$

thus $\ell(x(t), z(t))$ is non-increasing with respect to $t$, which proves Statement 1.

For Statement 2, note that

$$P(x)\nabla f(x) = \nabla f(x) + J_h(x)^\top \bar{\lambda}(x),$$

thus from (40) we have

$$\int_{t=0}^{T} \|z(t)\|^2 + \frac{a_2}{8} h(x(t))^\top K h(x(t)) + \frac{a_1}{4} \|P(x(t))\nabla f(x(t))\|^2 dt \leq \ell(x(0), z(0)) - \ell(x(T), z(T))$$
$$\implies \int_{t=0}^{T} \frac{a_2 \lambda_{\min}(K)}{8} \|h(x(t))\|^2 + \frac{a_1}{4} \|\nabla f(x(t)) + J_h(x(t))\bar{\lambda}(x(t))\|^2 dt \leq \ell(x(0), z(0)) - \ell_{\min}$$

which completes the proof.

We now prove Statement 3. Note that

$$\min\left\{ \frac{a_2 \lambda_{\min}(K)}{8}, \frac{a_1}{4} \right\} \texttt{KKT-gap}(x, \bar{\lambda}(x))^2 \leq \frac{a_2 \lambda_{\min}(K)}{8} \|h(x(t))\|^2 + \frac{a_1}{4} \|\nabla f(x(t)) + J_h(x(t))\bar{\lambda}(x(t))\|^2,$$

thus we have

$$\int_{t=0}^{T} \texttt{KKT-gap}(x(t), \bar{\lambda}(x(t)))^2 \leq \frac{\ell(x(0), z(0)) - \ell_{\min}}{\min\left\{ \frac{a_2 \lambda_{\min}(K)}{8}, \frac{a_1}{4} \right\}}$$

and thus gives

$$\inf_{0 \leq t \leq T} \texttt{KKT-gap}(x(t), \bar{\lambda}(x(t))) \leq \sqrt{\frac{\ell(x(0), z(0)) - \ell_{\min}}{\min\left\{ \frac{a_2 \lambda_{\min}(K)}{8}, \frac{a_1}{4} \right\} T}} \sim O(\frac{1}{\sqrt{T}})$$

and that

$$\lim_{t \to +\infty} \texttt{KKT-gap}(x(t), \bar{\lambda}(x(t))) = 0,$$

further, given that $\lim_{t \to +\infty} \bar{\lambda}(x(t)) - \lambda(t) = 0$, we have

$$\lim_{t \to +\infty} \texttt{KKT-gap}(x(t), \lambda(t)) = 0,$$

which completes the proof.

### F.1 Intuition: optimization with affine constraints

The main goal of this section is to provide intuition for why FL-momentum is able to accelerate convergence compared with FL-proximal. We will focus on the setting where the constraints are affine functions and show that in this setting, the FL-proximal method corresponds to the projected gradient descent, and the FL-momentum corresponds to its momentum-accelerated version.

The problem that we consider is

$$\min_x f(x)$$
$$s.t. \ \ Ax + b = 0, \tag{41}$$

In this setting, the forward Euler discretization for FL-proximal is given by

$$
\begin{aligned}
x_{t+1} &= x_t + \eta \nabla f(x_t) - \eta A^\top (AA^\top)^{-1} \left( A\nabla f(x_t) - \frac{1}{\eta}(Ax_t + b) \right) \\
&= (I - A^\top (AA^\top)^{-1} A)(x_t - \eta \nabla f(x_t)) - A^\top (AA^\top)^{-1} b \\
&= \mathrm{Proj}(x_t - \eta \nabla f(x_t)), \tag{42}
\end{aligned}
$$

where Proj is the projection onto the hyperplane $Ax + b = 0$. Note that the above scheme is equivalent to projected gradient descent. Further, the forward Euler discretization for FL-momentum is given by

$$
\begin{aligned}
w_t &= x_t + \beta(x_t - x_{t-1}) \\
\lambda_t &= - \left( J_h(w_t) J_h(w_t)^\top \right)^{-1} \left( J_h(w_t)\nabla f(w_t) - \frac{1}{\eta} h(w_t) \right) \\
x_{t+1} &= w_t + \eta \nabla f(w_t) - \eta A^\top (AA^\top)^{-1} \left( A\nabla f(w_t) - \frac{1}{\eta}(Aw_t + b) \right) \\
&= (I - A^\top (AA^\top)^{-1} A)(w_t - \eta \nabla f(w_t)) - A^\top (AA^\top)^{-1} b \\
&= \mathrm{Proj}(w_t - \eta \nabla f(w_t)),
\end{aligned}
$$

Note that this is exactly the momentum-accelerated projected gradient descent (cf. [20]). $\qquad\square$

## G   Auxiliaries

**Lemma 2.** *The following two problems are the dual problem for each other.*

$$\min_x \nabla f(x_t)^\top (x - x_t) + \frac{1}{2\eta}(x - x_t)^\top T(x_t)^{-1}(x - x_t)$$
$$s.t. \ \ h(x_t) + J_h(x_t)(x - x_t) \leq 0 \tag{43}$$

$$\min_{\lambda \geq 0} \ \frac{1}{2}\lambda^\top J_h(x_t)T(x_t)J_h(x_t)^\top \lambda + \lambda^\top \left( J_h(x_t)T(x_t)\nabla f(x_t) - \frac{1}{\eta} h(x_t) \right) \tag{44}$$

*Proof.* We write out the lagrangian for (43)

$$L(x, \lambda) = \nabla f(x_t)^\top (x - x_t) + \frac{1}{2\eta}(x - x_t)^\top T(x_t)^{-1}(x - x_t) + \lambda^\top \left( h(x_t) + J_h(x_t)(x - x_t) \right)$$

Note that strong duality holds for (43). Thus solving (43) is equivalent to solving

$$\min_x \max_{\lambda \geq 0} \nabla f(x_t)^\top (x - x_t) + \frac{1}{2\eta}(x - x_t)^\top T(x_t)^{-1}(x - x_t) + \lambda^\top \left( h(x_t) + J_h(x_t)(x - x_t) \right)$$
$$= \max_{\lambda \geq 0} \min_x \nabla f(x_t)^\top (x - x_t) + \frac{1}{2\eta}(x - x_t)^\top T(x_t)^{-1}(x - x_t) + \lambda^\top \left( h(x_t) + J_h(x_t)(x - x_t) \right)$$

Note that

$$\arg\min_x \nabla f(x_t)^\top (x - x_t) + \frac{1}{2}(x - x_t)^\top T(x_t)^{-1}(x - x_t) + \lambda^\top \left( h(x_t) + J_h(x_t)(x - x_t) \right)$$
$$= x_t - \eta T(x_t)(\nabla f(x_t) + J_h(x_t)^\top \lambda),$$

substituting this to the lagrangian we get

$$\max_{\lambda \geq 0} \min_x L(x, \lambda)$$

$$= \eta \left( \max_{\lambda \geq 0} -\nabla f(x)^\top T(x_t)(\nabla f(x_t) + J_h(x_t)^\top \lambda) + \frac{1}{2}(\nabla f(x_t) + J_h(x_t)^\top \lambda)^\top T(x_t)(\nabla f(x_t) + J_h(x_t)^\top \lambda) \right.$$

$$\left. + \lambda^\top \left( \frac{1}{\eta} h(x_t) + J_h(x_t)T(x_t)(\nabla f(x_t) - J_h(x_t)^\top \lambda) \right) \right)$$

$$= \eta \left( \max_{\lambda \geq 0} -\frac{1}{2}\lambda^\top J_h(x_t)T(x_t)J_h(x_t)^\top \lambda - \lambda^\top \left( J_h(x_t)T(x_t)\nabla f(x_t) - \frac{1}{\eta}h(x_t) \right) \right)$$

$$= -\eta \left( \min_{\lambda \geq 0} \frac{1}{2}\lambda^\top J_h(x_t)T(x_t)J_h(x_t)^\top \lambda + \lambda^\top \left( J_h(x_t)T(x_t)\nabla f(x_t) - \frac{1}{\eta}h(x_t) \right) \right)$$

The proof is thus completed by strong duality $\qquad \square$

**Lemma 3.** *Assumption 3 along with Assumption 1 is a sufficient condition of Assumption 4.*

*Proof.* From Lemma 1 we have that

$$\lambda^\top J_h(x)T(x)J_h(x)^\top \lambda + \lambda^\top (J_h(x)T(x)\nabla f(x) - Ky) = 0$$
$$\implies \lambda_{\min}(J_h(x)J_h(x)^\top)\lambda_{\min}\|\lambda^\top\| \leq \lambda^\top J_h(x)T(x)J_h(x)^\top \lambda$$
$$= -\lambda^\top(J_h(x)T(x)\nabla f(x) - Ky) \leq \|\lambda\|\|(J_h(x)T(x)\nabla f(x) - Ky)\|$$
$$\implies \|\lambda\| \leq \frac{\|J_h(x)T(x)\nabla f(x) - Ky\|}{\lambda_{\min}(J_h(x)J_h(x)^\top)\lambda_{\min}} \leq \frac{\lambda_{\max}\|(J_h(x)\nabla f(x)\| + \|\lambda_{\max}(K)h(x(0))\|}{\lambda_{\min}(J_h(x)J_h(x)^\top)\lambda_{\min}}$$

From Assumption 3 and Assumption 1 we have that

$$\frac{1}{\lambda_{\min}(J_h(x)J_h(x)^\top)} \leq D^2, \|(J_h(x)\nabla f(x)\| \leq M^2$$

And thus

$$\|\lambda\| \leq \frac{\lambda_{\max}M^2 + \|\lambda_{\max}(K)h(x(0))\|}{\lambda_{\min}}D^2.$$

Hence, Assumption 4 is satisfied by setting $L = \frac{\lambda_{\max}M^2 + \|\lambda_{\max}(K)h(x(0))\|}{\lambda_{\min}}D^2$. $\qquad \square$

