# OpenReview forum: "Constrained Optimization From a Control Perspective via Feedback Linearization"
_NeurIPS.cc/2025/Conference — NeurIPS 2025 poster_

### Official Review · Reviewer_SLKh · 2025-07-01

**Clarity:** 3
**Significance:** 3
**Originality:** 3
**Rating:** 4
**Confidence:** 4

**Summary:**

This paper introduces a method for solving constrained optimization problems using feedback linearization (FL), a concept from control theory. The authors establish global convergence rates for this method in equality-constrained problems and reveal a strong connection between the FL approach and the SQP algorithm. They then extend the method to handle inequality constraints and propose a momentum-accelerated FL algorithm that demonstrates faster convergence in practice. The theoretical claims are supported by proofs in the appendices and validated through numerical simulations on a logistic regression problem with fairness constraints.

**Questions:**

1. In nonlinear control, the feedback linearization control law may not be globally valid. How's this issue resolved?

2. The feedback linearization process often involves complex coordinate transformations and deriving the feedback control law. What is the computational complexity of the FL implementation?

3. How does this approach compare against other solvers? What are potential applications?

**Ethical Concerns:**

["NO or VERY MINOR ethics concerns only"]

**Final Justification:**

The pathway demonstrated by the author(s) to the global last-iterate guarantee seems promising and I hope this could be seriously addressed in the final manuscript. I've updated my score to 4.

**Limitations:**

The author(s) haven't discussed their limitations in the conclusion section.

1. The theoretical guarantees rely on strong conditions, such as the global boundedness of the objective and constraint function gradients (Assumption 1) and the universal invertibility of the $J_h(x)J_h(x)^\top$ matrix (Assumption 3). These assumptions may not hold in all practical scenarios.

2. While the FL-momentum method shows empirical speed-up, the theoretical analysis does not formally prove an accelerated convergence rate for the general nonconvex case; it provides the same $O(1/\sqrt{T})$ rate as the non-accelerated method.

3. The method's reliance on inverting an $m \times m$ matrix, where $m$ is the number of constraints, could be a bottleneck for problems with a very large number of constraints.

4. The analysis guarantees convergence for the best-found solution over time, which is a weaker guarantee than ensuring the final solution is optimal.

I'd like to change my rating if the author(s) would like to comment on the limitations above,

**Quality:**

3

**Strengths And Weaknesses:**

Strengths:

This is a solid theoretical paper that bridges the fields of nonlinear control theory and optimization. The most interesting contribution is the link established between feedback linearization and SQP, which provides a fresh perspective on a classic optimization algorithm and lends credibility to the FL approach. The development of a momentum-accelerated version is a practical and novel addition.

Weaknesses:

Despite its strengths, the paper has some limitations, as summarized below. Also, there is a missing 'n' in the paper title, and there might be more typos and mistakes in the main paper.

---

> ### Author Rebuttal · Authors · 2025-07-31
>
> We sincerely thank the reviewer for the constructive feedback and valuable comments! We also appreciate the careful reading that brought several typos to our attention. We will carefully proofread the manuscript and correct these issues in the revised version. Below, we address the reviewer’s technical comments one by one.
>
> 1. **Globally validness of feedback linearization**: *“In nonlinear control, the feedback linearization control law may not be globally valid. How's this issue resolved?”*
>
> **Response**: We appreciate the reviewer’s question. In case there’s any potential ambiguity, we would like to briefly clarify that FL is a nonlinear control technique that transforms a nonlinear system into an equivalent linear system via an exact change of variables. Unlike standard linearization, which relies on first-order approximations valid only near a nominal point, FL yields an exact transformation that can, under suitable conditions, hold over the entire domain of interest. However, this global validity depends on structural properties of the system, specifically the regularity of the transformation. If these conditions are not met, the transformation may be only locally valid or ill-defined.
>
> In our setting, Assumption 3 (LICQ) ensures these conditions are met: it guarantees that the constraint Jacobian has full row rank throughout the feasible domain, thereby ensuring the regularity of the transformation and the existence of a well-defined feedback linearization. We will revise the manuscript to clarify this connection and make the role of LICQ in ensuring validity more explicit.
>
> 2. **Computational complexity**: *In Question 2 and Limitation 3, the reviewer questions about the computational complexity of feedback linearization, as it involves computation of the inverse of a $m\times m$ matrix.*
>
> **Response**: We thank the reviewer for this valuable question. We acknowledge that computing the inverse of an $m \times m$ matrix is an inherent part of the feedback linearization (FL) approach. However, as noted in **Remark 1**, the method is particularly well-suited for settings where the number of constraints is significantly smaller than the dimension of the variable $x$—a common scenario in many practical applications, such as fair or safe learning, where only a small number of fairness or safety constraints need to be enforced.
>
> Moreover, even in cases where the number of constraints is large and the matrix inversion becomes computationally burdensome, we show that it is possible to **approximate** the inverse efficiently while still maintaining convergence guarantees. Specifically, in **Appendix B.2**, we present a modified FL algorithm that incorporates a **Proportional-Integral (PI) controller** to tolerate approximation error and still ensure convergence to an optimal solution. In the revised manuscript, we will add a brief discussion of this extension in Remark 1 to clarify this potential for reducing computational complexity.
>
> 3. **Comparison with existing methods and practical applications**: *“How does this approach compare against other solvers? What are potential applications?”*
>
> **Response**: With regard to comparison with existing methods, due to space limits, we have deferred a more thorough discussion and comparison in the appendix. We would like to kindly point the reviewer to **Remark 4** in the main text for a summary of the comparison, and **Appendix A** for a detailed comparison with existing first order methods, as well as **Appendix B.1** for corresponding numerical results to compare these methods. Here we quote Remark 4 for the convenience of the reviewer:
>
> >“We briefly compare FL-proximal methods with other first-order approaches, including Primal-Dual Gradient Descent (PDGD), Projected Gradient Descent (PGD), and the Augmented Lagrangian Method (ALM). PDGD has been well studied (Kose 1956, Wang and Elia 2011, Qu and Li 2018), but is largely limited to convex settings and may fail in nonconvex problems (Zhang and Luo 2020, Applegate et al. 2024). PGD also struggles with nonconvexity, as projections onto nonconvex sets are often intractable. ALM can handle nonconvex constraints, but each iteration requires solving a potentially expensive nonconvex subproblem. In contrast, when the constraint dimension is small relative to that of x, SQP-based methods like ours often perform better in practice (​​Gould et al. 2003, Liu et al. 2018)”
>
> With regard to potential applications, we would like to highlight that many important problems—such as fair learning, safe learning, heterogeneous federated learning, and optimal power flow—naturally fall into the category of constrained optimization. Our algorithm is well-suited for these settings and we have empirically evaluated the performance of our method in two representative domains: heterogeneous machine learning (see **Section 6**) and optimal power flow (see **Appendix B.3**).
>
> We hope the above explanation addresses the reviewer's concerns, and we would be glad to provide further clarification or engage in any additional discussions.
>
> 4. **Restrictive Assumptions**: *The reviewer found the Assumption 1 (Boundedness of Gradient) and Assumption 3 (LICQ) relatively restricted and may not hold in practical applications.*
>
> **Response**: We thank the reviewer for raising this important concern. As noted in Footnote 1, Assumption 1 only needs to hold within a compact region where the optimization variables reside during the algorithm’s execution. This is a commonly adopted assumption in the optimization literature (cf. Nocedal and Wright, Numerical Optimization).
>
> Regarding Assumption 3 (LICQ), we acknowledge that it is a strong condition; however, it is a standard assumption in constrained optimization. We provide further context and justification in Appendix A, under the paragraph labeled “SQP.” Moreover, in the inequality-constrained setting, our main convergence results rely on Assumption 4, which is strictly weaker than LICQ. We clarify this relationship in the discussion following Assumption 4 and provide a formal proof in Appendix G. While we recognize that these assumptions may not hold universally in practice, we include a dedicated discussion in Appendix A to highlight their limitations and the potential for extensions beyond the current setting.
>
> 5. **Convergence Rate for momentum acceleration**: *“While the FL-momentum method shows empirical speed-up, the theoretical analysis does not formally prove an accelerated convergence rate for the general nonconvex case; it provides the same $O(1/\sqrt{T})$ rate as the non-accelerated method.”*
>
> **Response**:  We thank the reviewer for this helpful observation. We would like to note that the feedback-linearization-based algorithms in this paper are primarily designed for nonconvex constrained optimization, where standard first-order methods—including momentum-based variants—are typically limited to a rate of $O(1/\sqrt{T})$. This is the rate we establish for FL-momentum as well. The theoretical result is not aimed at demonstrating acceleration, but rather at establishing the algorithm’s validity through rigorous analysis of its stability and convergence. Furthermore, although our primary focus is on the nonconvex setting, we also show in **Remark 5** and **Appendix F.1** that FL-momentum recovers the behavior of classical **momentum-accelerated projected gradient** methods in the convex case. This connection provides additional theoretical justification for our approach.
>
> 6. **Best-iterate instead of last-iterate convergence**: *“The analysis guarantees convergence for the best-found solution over time, which is a weaker guarantee than ensuring the final solution is optimal.”*
>
> **Response**: We fully acknowledge that our convergence analysis guarantees only the best-found solution over time, rather than optimality of the final iterate. This limitation is explicitly noted in our current manuscript (see Lines 162–164). We would also like to point out that convergence guarantees based on the best-found iterate are widely adopted in research on non-convex optimization (cf. Eq. (2) in Danilova et al., “Recent Theoretical Advances in Non-Convex Optimization,” 2022). Ensuring the optimality of the final iterate is substantially more challenging in non-convex landscapes, where saddle points may block convergence of the last iterate. We are grateful for the reviewer’s feedback and would be happy to explore potential extensions that could strengthen guarantees for the last iterate, particularly in convex settings, in future work.
>
> 7. **Lack of limitation statement in conclusion section.**
>
> **Response**: We thank the reviewer for highlighting the absence of an explicit summary of limitations in the conclusion section, and we sincerely apologize for this oversight. As noted in our responses above, several parts of the main text (e.g. Remark 5 and Lines 147, 162–164 and 245–249) already discuss the assumptions and limitations of our results. In response to the reviewer’s suggestion, we will add the following dedicated paragraph in the conclusion of the revised manuscript to more thoroughly summarize these limitations:
>
> >“Several limitations of our framework remain. First, our analysis ensures convergence of the best-found iterate to a first-order KKT point, but does not distinguish between local optima, global optima, and saddle points. Extending the framework to guarantee convergence to local optima via saddle-point escape mechanisms is an important direction for future work. Second, the applicability of feedback linearization depends on structural assumptions such as the Linear Independence Constraint Qualification (LICQ). Third, while FL-momentum empirically accelerates convergence, our current analysis yields the same $O(1/\sqrt{T})$ rate as its non-accelerated counterpart. Relaxing these assumptions and strengthening convergence guarantees remain promising directions for future research.”

---

> > ### Comment · Reviewer_SLKh · 2025-08-05
> >
> > Thank you for the clarification and referring the numerical results. Since there's an obvious typo in the title of the paper, I didn't delve deeper enough to the supplementary materials during the review phase. I will take them into account in the later discussion phase.
> >
> > The fact that the proposed algorithm reduces to momentum-accelerated projected gradient methods in the convex case is interesting. I hope the paper can be further strengthened by improving the current best-iterate guarantee.

---

> ### Author Response · Authors · 2025-08-05
> **Follow-Up Clarifications and Additional Last-Iterate Result**
>
> Thank you very much for your thoughtful follow-up and for taking the time to revisit the supplementary materials. We truly appreciate your engagement with the paper and your recognition of the connection to momentum-accelerated projected gradient methods in the convex setting.
>
> We would like to sincerely apologize for the typo in the title and appreciate you bringing it to our attention. We have since conducted a careful proofreading of the manuscript and corrected this and other minor issues to improve overall clarity and presentation.
>
> We also value your suggestion regarding improving the best-iterate guarantee. As we have mentioned in our response, in the nonconvex setting, the best-iterate guarantee is a widely adopted standard in nonconvex optimization literature (c.f. in Eq. (2) in *Danilova et al., “Recent Theoretical Advances in Non-Convex Optimization,” 2022*, Theorem 5.3 in *Carmon et al “Accelerated Methods for Non-Convex Optimization,” 2018*, Theorem 5 in *Agarwal et al. “On the Theory of Policy Gradient Methods,” 2021*). Improving upon the best iterate guarantee is a meaningful and challenging direction, and we hope to explore ways to strengthen our theoretical results in future work, including incorporating second-order information and saddle-point escape techniques.
>
> To supplement our earlier response, we would also like to add that it is possible to establish **global last-iterate guarantee** for FL-proximal algorithm (for Equality-Constrained Optimization) in the strongly convex setting (where $f$ is $\mu$-strongly convex and $h$ is affine). In this setting, we can show that
> $$f(x(t)) - f(x^\star) \le O(e^{-2\mu t}),  $$
> for $K = 2\mu I$. The proof closely follows the argument in the proof of Theorem 1, with a key modification: using the **strong convexity** of $f$, combined with the **fourth line of the equation at line 1129 in Appendix C**, we can derive that $$\frac{df}{dt} \le -2 \mu (f(x(t)) - f(x^\star)).----(\*) $$ We have included a brief proof of claim (\*) at the end of the discussion. While this result is not the main focus of the current paper, we believe it partially addresses your concern and we will include it in the final version of the manuscript. We are grateful to the reviewer for this valuable suggestion, which enhances the overall contribution of the paper.
>
> We hope that the above discussions help to clarify the contributions and address the concerns outlined in the initial review. If you feel that these clarifications and planned revisions resolve the key issues, we would be truly grateful if you would consider updating your score. In the meantime, please don’t hesitate to let us know if there is anything else we can further clarify during the discussion phase.
> Thank you again for your time, constructive feedback, and willingness to engage in the discussion!
>
> &nbsp;
> &nbsp;
> &nbsp;
> &nbsp;
> ***
> **Brief proof of the claim (\*):**
>
> Here we assume $h(x) = Ax + b$, we further denote the projection onto span of $A$ as $P_A := A^\top (AA^\top)^{-1}A$, and $P_A^\perp = I - P_A$. Then the key argument is that according to strong convexity
>
> \begin{align}
> f(x) - f(x^\star)\le \nabla f(x)^\top (x-x^\star) - \frac{\mu}{2}||x-x^\star||^2 \le \nabla f(x)^\top P_A^\perp (x-x^\star) -\frac{\mu}{2}||P_A^\perp(x-x^\star)||^2 + \nabla f(x)^\top P_A (x-x^\star) \\
> \le \frac{1}{2\mu}\nabla f(x)^\top P_A^\perp \nabla f(x) + \nabla f(x)^\top P_A (x-x^\star)= -\frac{1}{2\mu} \frac{df}{dt}
> \end{align}
> where the last equality is given by a simple algebraic manipulation of the fourth equation in Line 1129  of Appendix C, which proves the claim.

---

> > ### Comment · Reviewer_SLKh · 2025-08-06
> >
> > Thank you for the update. The pathway to global last-iterate guarantee seems promising and I hope this could be seriously addressed in the final manuscript. I've updated my score to 4.

---

> > > ### Author Response · Authors · 2025-08-06
> > >
> > > Thank you very much for your thoughtful engagement throughout the review and discussion process. We sincerely appreciate your willingness to revisit the paper in light of our clarifications, and we are grateful for the updated score. We will take your suggestion seriously and ensure that the final manuscript clearly presents the last-iterate guarantee and its implications. Thank you again for your time and effort in helping to improve the quality of the manuscript.

---

### Official Review · Reviewer_9MiY · 2025-07-02

**Clarity:** 4
**Significance:** 2
**Originality:** 3
**Rating:** 4
**Confidence:** 3

**Summary:**

This paper builds upon the idea proposed in [15], which uses feedback linearization as a continuous analog to solve constrained optimization problems. The authors establish a sublinear convergence rate for both equality- and inequality-constrained cases. They also draw a connection between the feedback linearization framework and sequential quadratic programming (SQP), showing that an Euler discretization of the feedback linearization setup is equivalent to SQP.

**Questions:**

What are some concrete application scenarios where this method would be particularly useful?

**Ethical Concerns:**

["NO or VERY MINOR ethics concerns only"]

**Final Justification:**

The response has addressed my concerns to a large extent. Overall, I believe this paper offers a valuable—albeit somewhat narrow—contribution, and I therefore recommend acceptance.

**Limitations:**

Yes.

**Quality:**

3

**Strengths And Weaknesses:**

Strengths:
- The convergence results contribute meaningfully to the existing literature on continuous-time optimization methods.
- The connection between feedback linearization and SQP is insightful. The design of the linear dynamics sheds light on the behavior of different SQP variants with different quadratic regularization terms.
- The inclusion of a momentum-accelerated version of the algorithm, along with a partial convergence result, is a nice addition.

Weaknesses:
- The relevance of this work to the NeurIPS audience is unclear. While the paper is well-suited for optimization, control, or applied mathematics venues, its connection to machine learning is limited.
- The momentum-accelerated variant has the same convergence rate as the original algorithm and performs worse empirically than the FL-Newton variant, making its practical value unclear.
- The paper presents only one numerical example, which limits the empirical validation of the approach.

---

> ### Author Rebuttal · Authors · 2025-07-31
>
> We sincerely thank the reviewer for the comments and feedback as well as the recognition of our theoretical contributions, the connection to SQP, and the momentum-based extension. Below we address the comments and concerns of the reviewer one by one:
>
> 1. **Relevance of the work to NeurIPS audiences**: *“While the paper is well-suited for optimization, control, or applied mathematics venues, its connection to machine learning is limited.”*
>
> **Response**: We sincerely thank the reviewer for the feedback. While our work is indeed grounded in control and optimization theory, we believe it contributes to a methodological area that continues to play a central role in machine learning. Optimization lies at the heart of nearly all ML methods: the training of models, from logistic regression to deep neural networks, relies on solving high-dimensional optimization problems. As emphasized in Bottou, Curtis, and Nocedal (2018, Optimization Methods and Software), many of the field’s most significant advances have been enabled by progress in optimization algorithms themselves.
>
> More specifically, constrained optimization has become increasingly central to several emerging areas in machine learning. In safety-critical domains, such as robotics (Alonso-Mora et al., IJRA 2017)) and power grids operations (Dommel and Tinney, PAS 2007), etc., machine learning models must satisfy hard safety or operational constraints at training and inference time. The need for optimization methods that offer convergence guarantees under constraints is critical in such settings, where violations can lead to safety breaches or system failure. In addition, fairness-aware machine learning imposes constraints to ensure statistical parity or mitigate disparate treatment. For instance, Agarwal et al. (ICML 2018) cast fair classification as a constrained optimization problem, showing how algorithmic tools can effectively balance accuracy and fairness criteria.
>
> Our contribution builds directly on this growing demand for principled constrained optimization methods in machine learning. By offering a scalable algorithm with theoretical convergence guarantees under general constraints, our work contributes to the foundational tools needed to support these applications. We will revise the manuscript to better highlight these connections and we appreciate the opportunity to clarify the relevance of our work. We hope this discussion addresses the reviewer’s concern, and we would be happy to further elaborate if helpful.
>
> 2. **Momentum acceleration rate**: *“The momentum-accelerated variant has the same convergence rate as the original algorithm and performs worse empirically than the FL-Newton variant, making its practical value unclear.”*
>
> **Response**: We thank the reviewer for raising this point. Regarding the reviewer’s observation about FL-momentum and FL-proximal sharing the same theoretical convergence rate, we note that FL-momentum is designed for nonconvex constrained optimization, where standard first-order methods—including momentum-based variants—are typically limited to a rate of $O(1/\sqrt{T})$. This is the rate we establish for FL-momentum as well. The theoretical result is not aimed at demonstrating acceleration, but rather at establishing the algorithm’s validity through rigorous analysis of its stability and convergence. Furthermore, although our primary focus is on the nonconvex setting, we also show in **Remark 5** and **Appendix F.1** that FL-momentum recovers the behavior of classical **momentum-accelerated projected gradient methods** in the convex case. This connection provides additional theoretical justification for our approach.
>
> Regarding the empirical comparison with FL-Newton, we agree that FL-Newton typically converges faster, as it leverages second-order (Hessian) information. In contrast, FL-momentum is a purely first-order method, which is expected to offer greater computational efficiency at the cost of slightly slower convergence than second-order methods. Our objective is not to outperform FL-Newton, but to show that FL-momentum provides a clear acceleration over FL-proximal, our first-order baseline. We include FL-Newton as a reference to illustrate that FL-momentum can achieve competitive performance while avoiding the computational overhead associated with second-order methods. We will clarify this distinction in the revised manuscript.
>
> 3. **Lack of numerical examples**: *“The paper presents only one numerical example, which limits the empirical validation of the approach.”*
>
> **Response**: We acknowledge the reviewer’s concern about the limited number of numerical examples. We would like to kindly point out that apart from the numerical simulations in Section 6, additional numerical experiments are also presented in **Appendix B**. In particular, Appendix B.1 compares our approach with other first-order constrained optimization algorithms and Appendix B.3 provides an additional numerical example on the **optimal power flow** problem, an important real-world constrained optimization problem. Due to space constraints, we have moved the full details to the appendix, but we have highlighted the setting and key results in the main text.
>
> 4. **Practical applications**: *“What are some concrete application scenarios where this method would be particularly useful?”*
>
> **Response**: We thank the reviewer for this question. As mentioned in response to the reviewer’s Concern 1, many important problems—such as safe or fair learning, heterogeneous federated learning, and optimal power flow—naturally fall into the category of constrained optimization, where enforcing constraints on model behavior or distributional fairness is critical.  Our approach is well-suited for these settings. To demonstrate its applicability, we have empirically evaluated our method in two representative domains: heterogeneous machine learning (see **Section 6**) and optimal power flow (see **Appendix B.3**).
>
> Once again, we sincerely thank the reviewer for the constructive feedback and for recognizing the contributions of our work. We hope that our responses have sufficiently addressed the concerns raised by the reviewer. We remain open to further discussion and would be glad to elaborate further if needed.

---

> > ### Comment · Reviewer_9MiY · 2025-08-06
> > **Post-rebuttal assessment**
> >
> > I would like to thank the authors for their rebuttal. The response has addressed my concerns to a large extent. Overall, I believe this paper offers a valuable—albeit somewhat narrow—contribution, and I therefore maintain my positive assessment of the work.

---

> > > ### Author Response · Authors · 2025-08-06
> > >
> > > Thank you very much for your feedback and for taking the time to review and engage with our work! We’re glad to hear that our response helped address your concerns, and we truly appreciate your positive assessment and support for the contribution. Your comments and suggestions have been very helpful in improving the clarity and presentation of the paper, and we will make sure to reflect them in the final version.

---

### Official Review · Reviewer_fiZS · 2025-07-02

**Clarity:** 3
**Significance:** 3
**Originality:** 3
**Rating:** 5
**Confidence:** 4

**Summary:**

The paper discusses methods for solving non-convex constrained optimization using a feedback linearization approach. Specifically, the optimization process is considered as a dynamical system, where the vector field is the first derivative of the Lagrangian, and the Lagrange multipliers are inputs to the system.   The feedback linearization approach allows the authors to formulate the control law in terms of the gradients and Jacobian matrix of the objective and constraints. Then, the stability of the closed-loop system implies convergence of the states to an equilibrium point with respect to the KKT-gap. Finally, discretization of the continuous-time system shows the equivalence of the feedback linearization approach to SQP algorithms. The second part of the paper generalizes the results to inequality constrained problems.

**Questions:**

I would have liked to see an analysis of how the proposed approach compares with traditional methods. The results are interesting from a theoretical perspective, but is this a better approach in general and will this line of research ultimately have a significant practical benefit?

**Ethical Concerns:**

["NO or VERY MINOR ethics concerns only"]

**Final Justification:**

The authors have prepared careful responses to reviewer questions. The paper is well-written and the results interesting. I have no significant concerns.

**Limitations:**

no limitations acknowledged.

**Paper Formatting Concerns:**

Overuse of remark environment. Literature placement in appendix made understanding advancement over state of the art more difficult.

**Quality:**

4

**Strengths And Weaknesses:**

In general, the paper is well-structured and the contribution is sufficient. However, there are several comments.

First, the global  convergence terminology for the paper is unusual. The authors consider the KKT-gap as a convergence metric, which should be addressed more carefully. For example, Theorems 1 and 4 show that the KKT gap goes to zero. However, the first-order KKT conditions (the KKT gap is equal $0$) are a necessary condition for optimality, so the equilibrium point $x*$ may not be a solution to the optimization problem.  Furthermore, the convergence of $x(t)$ to any equilibrium point is not guaranteed.  Thus, I would recommend adding a definition of global convergence as used in the theorems. Also, for clarity of the results, the paper should discuss the relationship between the global convergence and the convergence of $x(t)$ to the optimal solution $x*$.

Second, Lyapunov functions are required to be positive-definite everywhere except the equilibrium point, where the function must be equal $0$. However, the proposed Lyapunov functions, $l$, in theorems (Thm. 1 and 4) do not satisfy the condition. These conditions, along with non-increasing with respect to time of the Lyapunov function, guarantee the stability property of the system. Authors should avoid calling this function a Lyapunov function or modify it to satisfy the conditions.

Third,  the relationship with SQP methods is interesting. However, the assumptions required for Thms. 2 and 3 are missing. For example, Thm. 3 leverages strong duality. However, the optimization in Eqn. (15) is not necessarily convex, because of non-convexity of $f$ and $h$.

Finally, there are minor formatting issues. Specifically, equations in Property 3 in Theorems 3 and 4 should be formatted more accurately. For example, Theorem 3 the norm symbol in the equation is missing. The equation in Theorem~4 (Property 3) is hard to understand without additional spaces between symbols.

---

> ### Author Rebuttal · Authors · 2025-07-31
>
> We sincerely thank the reviewer for carefully reading our paper and providing constructive feedback, as well as for their positive assessment of our work! Below, we address the technical questions raised by the reviewer.
>
> 1. **The “global convergence” terminology**: *The reviewer points out that the use of “global convergence” in the paper is unconventional, as the theorems only show that the KKT gap converges to zero, which is a necessary but not sufficient condition for optimality.*
>
>  **Response**: We thank the reviewer for this important observation. Indeed in the paper we focus on convergence of the KKT gap to zero from any initial condition. We agree that this does not imply convergence to a global or local optimum, especially in the non-convex setting. Our goal is to establish convergence to a first-order KKT point, which, although an easier task than converging to the global optimal, is still challenging in the non-convex constrained optimization settings. We will revise the manuscript to clarify that our convergence criterion is based on KKT gap and acknowledge its limitations. The following discussion will be added:
> >“In this paper, we use the term ‘convergence’ to refer to the decay of the KKT gap to zero. While this is weaker than convergence to a local or global optimum, it remains a meaningful guarantee in nonconvex constrained optimization. Extending the framework to incorporate saddle-point escape techniques and ensure convergence to local optima is an important direction for future work.”
>
> 2. **The “Lyapunov function” terminology**: *The reviewer points out that the functions labeled as Lyapunov functions in Theorems 1 and 4 do not meet the standard definition (i.e., being positive-definite and zero only at the equilibrium)*.
>
> **Response**: We agree with the reviewer that the proposed function does not meet the strict criteria of a Lyapunov function. To avoid confusion, we will revise the terminology and refer to it as a "merit function", which is more appropriate in the context of optimization. We sincerely thank the reviewer for this helpful suggestion, which improves both the clarity and rigor of the paper.
>
> 3. **Assumptions for Theorems 2 and 3**: *The reviewer notes that some assumptions for Theorems 2 and 3 are missing. The reviewer also questions whether strong duality still holds in Theorem 3 since the objective function $f$ and constraints $h$ can be nonconvex.*
>
> **Response**: We thank the reviewer for the valuable feedback. For Theorem 2, we acknowledge that it relies on Assumption 3, which ensures the invertibility of $J_h(x)J_h(x)^\top$. This condition guarantees the validity of the feedback linearization and, accordingly, we will revise the manuscript to clearly state this requirement in the theorem statement.
>
> Regarding Theorem 3, we would like to clarify that the only necessary assumption is the feasibility of Equation (15), as noted in the paper. While it is true that $f$ and $h$ may be non-convex in the original problem, Theorem 3 concerns convex quadratic programming subproblems (Equation (14) and (15)), which are solved at each iteration. Under the relaxed Slater condition, strong duality holds for this subproblem even when the overall problem is non-convex. We have revised the text to emphasize this distinction and ensure the assumptions are stated with better clarity. We are grateful to the reviewer for prompting this important clarification and we would be happy to elaborate further if needed.
>
> 4. **Comparison with existing traditional methods and practical benefits**: *The reviewer finds the theoretical framework interesting but asks whether the proposed approach offers practical benefits.*
>
>
> **Response**: We thank the reviewer for finding the theoretical framework interesting and for raising the questions. With regard to comparison with existing methods, due to the space limits, we have deferred a more thorough discussion and comparison in the appendix. We would like to kindly point the reviewer to **Remark 4** in the main text for a summary of the comparison, and **Appendix A** for a detailed comparison with existing first order methods, as well as Appendix **B.1** for corresponding numerical results to compare these methods. Here we briefly quote Remark 4 for the convenience of the reviewer:
>
> >“We briefly compare FL-proximal methods with other first-order approaches, including Primal-Dual Gradient Descent (PDGD), Projected Gradient Descent (PGD), and the Augmented Lagrangian Method (ALM). PDGD has been well studied (Kose 1956, Wang and Elia 2011, Qu and Li 2018), but is largely limited to convex settings and may fail in nonconvex problems (Zhang and Luo 2020, Applegate et al. 2024). PGD also struggles with nonconvexity, as projections onto nonconvex sets are often intractable. ALM can handle nonconvex constraints, but each iteration requires solving a potentially expensive nonconvex subproblem. In contrast, when the constraint dimension is small relative to that of x, SQP-based methods like ours often perform better in practice (​​Gould et al. 2003, Liu et al. 2018)”
>
>
> With regard to the practical benefits of the proposed theoretical framework, we would like to emphasize that, while our perspective provides an alternative—but mathematically equivalent—formulation of the first-order KKT conditions through the lens of control and dynamical systems, it offers a fresh viewpoint that can inspire novel algorithmic designs with potentially advantageous properties. For example, our momentum-accelerated method is directly motivated by momentum mechanisms in dynamical systems. Furthermore, as discussed in **Remark 2 (Extensions of the FL approach)** and **Appendix B.2**, our framework opens up additional opportunities for control-inspired enhancements. In particular, beyond using purely proportional control, one could employ **Proportional-Integral (PI) control** to improve robustness against system misspecification or approximation errors in the computation of the inverse of $J_h(x)J_h(x)^\top$. To the best of our knowledge, such PI-inspired approaches have not been explored in the optimization literature, and we believe this also illustrates the broader potential of our methodology.
>
> We hope the above discussion helps clarify the reviewer’s questions, and we would be happy to engage in further discussion if needed.
>
> 5. **Formatting issues.**
>
> **Response**: We sincerely thank the reviewer for reading through the technical sections of the papers carefully and thoroughly! We’ve corrected the formatting issues raised by the reviewer in the revised version of the paper.
>
> 6. **No limitation acknowledged.**
>
> **Response**: We thank the reviewer for pointing out the absence of an explicit summary of limitations in the conclusion section. We sincerely apologize for this oversight. That said, we would like to respectfully note that several parts of the main text (such as Remark 5 and Lines 147 and 245–249) do discuss the assumptions and limitations of our results. In response to the reviewer’s suggestion, we will add the following dedicated paragraph in the conclusion of the revised manuscript to more thoroughly summarize these limitations:
> >“Several limitations of our framework remain. First, our analysis ensures convergence of the best-found iterate to a first-order KKT point, but does not distinguish between local optima, global optima, and saddle points. Extending the framework to guarantee convergence to local optima via saddle-point escape mechanisms is an important direction for future work. Second, the applicability of feedback linearization depends on structural assumptions such as the Linear Independence Constraint Qualification (LICQ). Third, while FL-momentum empirically accelerates convergence, our current analysis yields the same $O(1/\sqrt{T})$ rate as its non-accelerated counterpart. Relaxing these assumptions and strengthening convergence guarantees remain promising directions for future research.”
>
> We sincerely thank the reviewer once again for the thoughtful and careful reading of our paper, and for recognizing its contributions. We greatly appreciate the time and effort dedicated to understanding our work in depth. We are encouraged by the feedback and would be happy to engage in further discussions or clarifications.

---

### Author Response · Authors · 2025-08-04
**Clarifications and Follow-Up Ahead of Discussion Deadline**

We sincerely appreciate the time and valuable feedback provided by all reviewers. We have carefully addressed each of your comments in our rebuttal and will revise the manuscript accordingly to improve clarity, precision, and rigor. If there are any remaining concerns or points that would benefit from further clarification, we would be happy to discuss them.

With the discussion deadline approaching, we would like to express our appreciation once again and hope that our responses and planned revisions help clarify the value and relevance of our contributions. We would be grateful if you would kindly consider acknowledging the response—and we sincerely thank the reviewer(s) who have already done so—and updating your score if you feel the concerns have been adequately addressed.

Thank you again for your time and constructive input!

---

### Decision · Program_Chairs · 2025-09-17

**Decision:**

Accept (poster)

**Comment:**

This paper builds on the idea of reference [15], which first proposed applying feedback linearization (FL) to constrained optimization, and substantially extends it in both scope and depth. The authors show that modeling optimization dynamics in continuous time and applying FL leads to convergence to first-order KKT points. A key contribution is establishing the connection between FL and Sequential Quadratic Programming (SQP), where discretization of the FL dynamics recovers SQP updates. The framework is extended from equality- to inequality-constrained problems, and a momentum-accelerated variant is proposed with theoretical guarantees and supporting experiments. Overall, the paper bridges control and optimization, providing new perspectives on classical constrained methods.

Reviewers agreed the work is rigorous and well-structured, highlighting the SQP connection and momentum-based extension as notable contributions. Concerns centered on restrictive conditions (bounded gradients, LICQ), limited empirical validation, and unclear ML relevance. In rebuttal, the authors constructively clarified terminology (introducing “merit function”), explicitly added missing assumptions, emphasized broader ML applications such as safe/fair learning, federated optimization, and power flow, and acknowledged limitations directly in the conclusion. They also pointed to additional experiments in the appendix and clarified the role of FL-momentum relative to FL-Newton.

Overall, this work provides further novel perspective on constrained optimization through the lens of feedback linearization. While empirical validation remains modest and the assumptions limit generality, the methodological contribution and relevance to constrained ML applications justify acceptance.